# LENGTH-INDUCED EMBEDDING COLLAPSE IN TRANSFORMER-BASED MODELS

## ABSTRACT

Text embeddings enable various applications, but their performance deteriorates on longer texts. In this paper, we find that the performance degradation is due to a phenomenon called **Length Collapse**, where longer text embeddings collapse into a narrow space. This collapse results in a distributional inconsistency between embeddings of different text lengths, ultimately hurting the performance of downstream tasks. Theoretically, by considering the self-attention mechanism inherently functions as a low-pass filter, we prove that long sequences increase the attenuation rate of the low-pass filter effect of the self-attention mechanism. With layers going deeper, excessive low-pass filtering causes the token signals to retain only their Direct-Current (DC) component, which means the input token feature maps will collapse into a narrow space, especially in long texts. Based on the above analysis, we propose to mitigate the undesirable length collapse limitation by introducing a temperature in $\mathrm{softmax}(\cdot)$, which achieves a higher low-filter attenuation rate. The tuning-free method, called **TempScale**, can be plugged into multiple transformer-based embedding models. Empirically, we demonstrate that TempScale can improve existing embedding models especially on long text inputs, bringing up to **0.53%** performance gains on 40 datasets from Massive Text Embedding Benchmark (MTEB) and **0.82%** performance gains on 4 datasets from LongEmbed, which specifically focuses on long context retrieval. The source code is available at https://anonymous.4open.science/r/Length_Collapse-22D2.

## 1 INTRODUCTION

Text embeddings—dense vectors that preserve the semantic information of given texts—have become fundamental to many downstream natural language processing (NLP) applications, including text analysis (Aggarwal & Zhai, 2012; Angelov, 2020), question answering (Tan et al., 2023; Xu et al., 2024), web search (Zhao et al., 2023; Yates et al., 2021), and retrieval-augmented generation (Gao et al., 2023; Fan et al., 2024). Typically, embeddings are generated by pre-trained language models (PLMs), which produce fixed-dimensional embeddings regardless of the input text length. In practice, we expect PLMs to perform consistently on texts of varying lengths in any downstream applications.

Unfortunately, we observe that *popular transformer-based embedding models perform poorly on longer texts*. As shown in Figure 1a, using the classification task on the IMDB dataset from the Massive Text Embedding Benchmark (MTEB) (Muennighoff et al., 2023) leaderboard as an example, we evaluate the performance of mainstream embedding models on test sets grouped by different text lengths. The experimental results reveal that models of different capabilities and context window sizes consistently exhibit performance degradation as text length increases. For instance, the BGE (Xiao et al., 2023) model's classification accuracy drops significantly from 75.6% in the length range [0, 100) tokens to 59.0% in the range [400, 500) tokens, indicating a substantial decline of 16.6% points.

We attribute this performance degradation to a biased behavior of embedding models: embeddings of longer texts tend to cluster together, a phenomenon we term as **length collapse**. To verify this, we conduct controlled experiments depicted in Figures 1b and 1c. Figure 1b shows that embeddings of longer texts are more densely clustered near the origin in the dimensionally reduced embedding space, indicating a collapse that reduces variance among embeddings. Details of the rewriting process can be found in Appendix E.1. Figure 1c further demonstrates that embeddings of longer texts exhibit higher

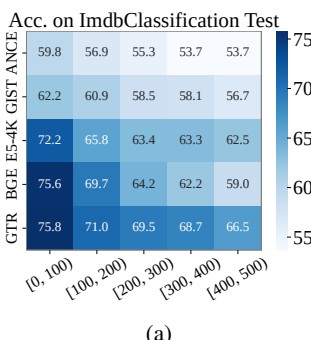 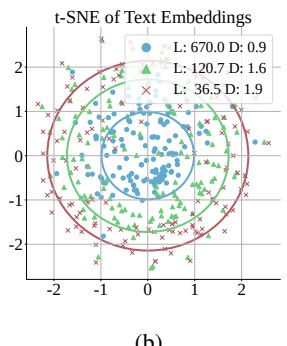 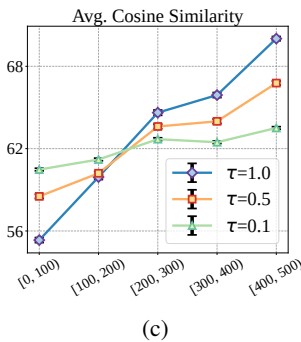

|     |     |     |
| --- | --- | --- |
| (a) | (b) | (c) |

Figure 1: **(a)** Performance of embedding models on IMDb classification across length intervals [0, 100) to [400, 500). The bluer ▇ a cell, the higher the classification accuracy. **(b)** t-SNE visualization of embeddings from the BGE on NFCorpus dataset, with ● for the original dataset and ▲ and × for LLM-summarized versions, retaining semantic meaning with varying lengths. **L** indicates average text length, and **D** denotes mean distance to the origin. **(c)** Mean pairwise cosine similarity of embeddings from the BGE model across text length intervals on the corpus from NFCorpus, with an X-axis for length intervals and a Y-axis for average pairwise similarity.

cosine similarity to each other, leading to smaller differences between them. This collapse results in a distributional inconsistency between embeddings of different text lengths, adversely affecting the performance of various downstream tasks (Yan et al., 2021) such as text classification.

To study how text length affects the distribution of embeddings, we conduct a rigorous analysis of the self-attention mechanism in Fourier space (Wang et al., 2022b) (Section 2.2). Based on the finding that cascading self-attention blocks are equivalent to repeatedly applying a low-pass filter, we further prove that the attenuation rate of the low-pass filter is proportional to the largest singular value $\sigma_a$ of the high-frequency components (HC) in self-attention matrix. Furthermore, assuming that the input keys and query tokens follow a Gaussian distribution, we can prove that the $\sigma_a$ value decreases as text length increases. As a result, longer input texts retain more of the Direct Component (DC) in the token signals, which means that the embeddings at the output layer will collapse to a very narrow space. This theoretical proof explains the observations in Figure 1b.

Building on this theoretical analysis, we propose a simple yet effective solution named Temperature Scaling (**TempScale**) to mitigate the length collapse phenomenon. TempScale manipulates the calculated attention map by dividing the attention scores by a parameter named temperature smaller than 1 before applying the $\mathrm{softmax}(\cdot)$ operator. This adjustment increases the variance of the attention score matrix, as shown in Eqn. 3, leading to a larger $\sigma_s$ for the self-attention matrix. As a result, the self-attention matrix has a lower filter attenuation rate, leading to more diverse embeddings. As shown in Figure 1c, a smaller temperature enables embeddings to exhibit lower pairwise cosine similarity and results in a more even distribution, thereby alleviating length collapse.

Our contributions can be summarized as follows:

- We uncover the **length collapse** phenomenon and then establish a rigorous theoretical analysis from the spectral domain and show that the length collapse is due to the low-pass filtering strength of self-attention increases as the sequence length increases, leading the token signals retain only their DC component.

- We present TempScale, a theoretically grounded scaling technique that incorporates a temperature parameter into the $\mathrm{softmax}(\cdot)$ function to achieve a higher attenuation rate in the self-attention matrix. TempScale is efficient, easy to use, tuning-free, and able to generalize across different embedding models compared to extending context window methods.

- We conduct extensive experiments by integrating TempScale with mainstream embedding models. We demonstrate that TempScale can improve existing embedding models especially on long text inputs, bringing up to **0.53%** performance gains on 40 different datasets from Massive Text Embedding Benchmark (MTEB) (Muennighoff et al., 2023) and **0.82%** performance gains on 4 tasks from LongEmbed (Zhu et al., 2024).

## 2 LONG INPUTS LEAD TO HOMOGENEOUS EMBEDDINGS

In this section, we first present our notations and define the problem. Then, we introduce the Transformer structure in the mainstream embedding model and briefly explain the Fourier transform used in (Wang et al., 2022b). Based on the Fourier transform, we show that the attention mechanism acts as a low-pass filter, and longer input sequences strengthen the filtering effect, leading to increasingly similar representations. This results in cosine similarity increasing with the length of the text.

### 2.1 PRELIMINARIES AND BACKGROUND

**Notations.** Let $\boldsymbol{X} \in \mathbb{R}^{n \times d}$ denote the input feature matrix, where $n$ is the number of input tokens, and $d$ is the embedding dimension. Let $\boldsymbol{x}_i \in \mathbb{R}^d$ represent the vector corresponding to the $i$-th token and $\boldsymbol{z}_j \in \mathbb{R}^n$ represent the token sequence corresponding to the $j$-th dimension, where $i \in \{1, \ldots, n\}$ denotes the $i$-th row, and $j \in \{1, \ldots, d\}$ denotes the $j$-th column.

**Transformer Architecture.** In most modern embedding models (Chen et al., 2024; Xiong et al., 2021), a bidirectional transformer architecture based on attention mechanisms is widely used. These models generally consist of three key components: the embedding layer, a stack of transformer encoder blocks incorporating Multi-Head Self-Attention (MSA) and Feed-Forward Networks (FFN), and a pooling layer at the end to generate the final embedding representation of the input sequence. The Self-Attention(SA) module is the fundamental part of MSA, which takes inputs consisting of the token representations $\boldsymbol{X}$ from the previous layer, and it encodes each token by aggregating information from other tokens based on the attention scores, formulated as below (Vaswani, 2017):

$$\mathrm{SA}(\boldsymbol{X}) = \mathrm{softmax}\left(\frac{\boldsymbol{X}\boldsymbol{W}_Q(\boldsymbol{X}\boldsymbol{W}_K)^T}{\sqrt{d}}\right)\boldsymbol{X}\boldsymbol{W}_V, \tag{1}$$

where $\boldsymbol{W}_K \in \mathbb{R}^{d \times d_k}$, $\boldsymbol{W}_Q \in \mathbb{R}^{d \times d_q}$, $\boldsymbol{W}_V \in \mathbb{R}^{d \times d}$ are the key, query and value weight matrices, respectively. The dimensions of the query and key vectors are denoted by $d_q$ and $d_k$, while $\sqrt{d}$ serves as a scaling factor to adjust the magnitude of the dot product. The function $\mathrm{softmax}(\cdot)$ normalizes the attention scores row-wisely. Multi-Head Self-Attention (MSA) consists of SA heads, with their outputs combined through a linear projection:

$$\mathrm{MSA}(\boldsymbol{X}) = [\mathrm{SA}_1(\boldsymbol{X}) \quad \cdots \quad \mathrm{SA}_H(\boldsymbol{X})]\,\boldsymbol{W}_O,$$

where the subscripts indicate the number of self-attention (SA) heads, $H$ represents the total number of heads, and $\boldsymbol{W}_O \in \mathbb{R}^{Hd \times d}$ projects the combined multi-head outputs back to the hidden dimension.

**Fourier Analysis.** We use Fourier transform as the main analytic tool in this paper as used in (Wang et al., 2022b). Let $\mathcal{F} : \mathbb{R}^n \to \mathbb{C}^n$ represent the Discrete Fourier Transform (DFT), with its inverse, the Inverse Discrete Fourier Transform (IDFT), denoted by $\mathcal{F}^{-1} : \mathbb{C}^n \to \mathbb{R}^n$. Applying $\mathcal{F}$ to a token sequence $\boldsymbol{z}$ is equivalent to left multiplying a DFT matrix, where $k$-th row of DFT matrix denotes the Fourier basis corresponding to a certain frequency $\boldsymbol{f}_k = \begin{bmatrix} e^{2\pi j(k-1)\cdot 0} & \cdots & e^{2\pi j(k-1)\cdot(n-1)} \end{bmatrix}^\top / \sqrt{n} \in \mathbb{R}^n$, and $j$ is the imaginary unit. Let $\tilde{\boldsymbol{z}} = \mathcal{F}\boldsymbol{z}$ represent the spectrum of $\boldsymbol{z}$, where $\tilde{\boldsymbol{z}}_{dc} \in \mathbb{C}$ and $\tilde{\boldsymbol{z}}_{hc} \in \mathbb{C}^{n-1}$ correspond to the first element and the remaining elements of $\tilde{\boldsymbol{z}}$, respectively. We define the Direct-Current (DC) component of the input sequence $\boldsymbol{z}$ as $\mathcal{DC}[\boldsymbol{z}] = \tilde{\boldsymbol{z}}_{dc}\boldsymbol{f}_1 \in \mathbb{C}^n$, and the complementary high-frequency component as $\mathcal{HC}[\boldsymbol{z}] = \begin{bmatrix} \boldsymbol{f}_2 & \cdots & \boldsymbol{f}_n \end{bmatrix} \tilde{\boldsymbol{z}}_{hc} \in \mathbb{C}^n$, consistent with the definition in (Wang et al., 2022b).

In signal processing, a low-pass filter is a system that attenuates the high-frequency components of a signal while retaining the low-frequency components. In this paper, we specifically define a low-pass filter as one that preserves only the DC component $\mathcal{DC}[\boldsymbol{z}]$, while diminishing all other high-frequency components $\mathcal{HC}[\boldsymbol{z}]$. A more precise definition is provided in Definition 1.

**Definition 1.** *Let $f : \mathbb{R}^n \to \mathbb{R}^n$ be an endomorphism with $f^t$ obtained by applying $f$ for $t$ times. The function $f$ acts as a low-pass filter if and only if $\lim_{t \to \infty} \frac{\|\mathcal{HC}[f^t(\mathbf{z})]\|_2}{\|\mathcal{DC}[f^t(\mathbf{z})]\|_2} = 0$ for all $\boldsymbol{z} \in \mathbb{R}^n$.*

For additional background information, please refer to Appendix A.

### 2.2 THEORETICAL ANALYSIS ON LENGTH COLLAPSE

**Overview.** In this subsection, we aim to demonstrate that increasing the sequence length $n$ accelerates the rate of low-pass filtering in the attention matrix, leading to greater similarity in text embeddings

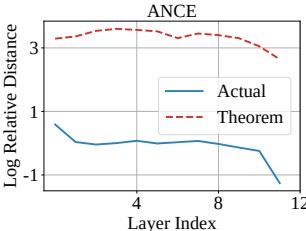 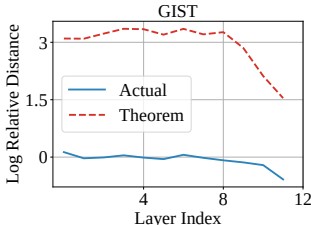 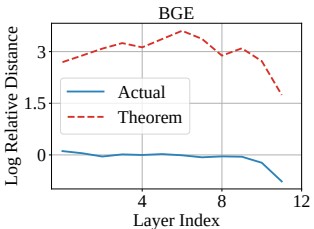

Figure 2: Visualization of the intensity of high-frequency components and their theoretical upper bounds. The blue line is defined by $\log(\|\mathcal{HC}[\boldsymbol{X}_{l+1}]\|_F / \|\mathcal{HC}[\boldsymbol{X}_l]\|_F)$, and the red line is estimated using the results in Theorem 2. More details can be found in Appendix E.2.

for longer texts, and provide a theoretical justification of self-attention based on its spectral-domain effect. Based on the previous findings that self-attention is constantly a low-pass filter (**Lemma** 1), which continuously erases high-frequency information, our main result is that the largest singular value $\sigma_a$ of $\mathcal{HC}[\boldsymbol{A}]$ can influence the filtering rate, with a smaller $\sigma_a$ indicating that self-attention can eliminate more high-frequency information (**Theorem** 2). Furthermore, our analysis shows that longer input sequences lead to a smaller $\sigma_a$ value for the attention matrix, resulting in the embedding of longer texts losing more feature expressiveness (**Theorem** 3). In conclusion, based on the assumption that natural language texts exhibit relatively consistent means, we can infer that longer texts tend to yield more similar representations (**Corollary 4**) and cause the **length collapse**. Lastly, we discuss the impact of other modules in Transformer on length collapse and distinguish our work from prior studies on similar collapse phenomena induced by deep layers in Transformers.

Formally, the following lemma demonstrates that the attention matrix generated by a softmax function (e.g., Eqn. 1) acts as a low-pass filter, independent of the specific token features or context window.

**Lemma 1.** *(Attention Matrix is A Low-pass Filter) Let $\boldsymbol{A} = \text{softmax}(\boldsymbol{P})$, where $\boldsymbol{P} \in \mathbb{R}^{n \times n}$. Then $\boldsymbol{A}$ must be a low-pass filter. For all $\boldsymbol{z} \in \mathbb{R}^n$,*

$$\lim_{t \to \infty} \frac{\|\mathcal{HC}[\boldsymbol{A}^t \boldsymbol{z}]\|_2}{\|\mathcal{DC}[\boldsymbol{A}^t \boldsymbol{z}]\|_2} = 0.$$

Lemma 1 follows directly from the Perron-Frobenius theorem (Meyer, 2000). Since all elements of the self-attention matrix are positive and each row sums to 1, the largest eigenvalue is 1. One can see that repeatedly applying the self-attention matrix can be viewed as the forward process of the embedding model. As the number of layers in the embedding model increases indefinitely, the final output retains only the DC component and thus loses all the feature expressive power. More detailed proofs of this lemma are provided in Theorem 1 in Wang et al. (2022b).

Understanding that self-attention matrices act as low-pass filters, we are interested in the extent to which an SA layer suppresses high-frequency components. Additionally, we provide a filter rate to illustrate the speed at which these high-frequency components are eliminated.

**Theorem 2.** *(Filter Rate of SA) Let $\sigma_a$ be the largest singular value of $\mathcal{HC}[\boldsymbol{A}]$ and $\text{SA}(\boldsymbol{X}) = \boldsymbol{A}\boldsymbol{X}\boldsymbol{W}_V$ the output of a self-attention module, we have*

$$\|\mathcal{HC}[\text{SA}(\boldsymbol{X})]\|_F \leq \sigma_a \|\boldsymbol{W}_V\|_2 \|\mathcal{HC}[\boldsymbol{X}]\|_F. \tag{2}$$

The proof of Theorem 2 can be found in Appendix B.1. Theorem 2 suggests the high-frequency intensity ratio to the pre- and post- attention aggregation is upper bounded by $\sigma_a \|\boldsymbol{W}_V\|_2$. When $\sigma_a \|\boldsymbol{W}_V\|_2 < 1$, $\mathcal{HC}[\boldsymbol{X}]$ converges to zero exponentially. We further present Figure 2 to justify our results, showing that the upper bound is consistent with the trend observed in the actual values.

Based on the proof that a lower $\sigma_s$ will lead to a higher filter-pass rate, we give proof that the $\sigma_s$ will decrease with input length $n$ increases in the following theorem.

**Theorem 3.** *(Filter Rate of Different Input Length $n$) Let $\boldsymbol{X}\boldsymbol{W}_Q$ and $\boldsymbol{X}\boldsymbol{W}_K$ be a Gaussian matrix, where elements $q_{ij} \sim \mathcal{N}(0, \sigma_q^2)$ and $k_{ij} \sim \mathcal{N}(0, \sigma_k^2), \forall i, j$. Let $p_{ij} = \boldsymbol{q}_i^\top \boldsymbol{k}_j / \sqrt{d}$ the attention*

*score of pair $i, j$, whose variance can be expressed as $\sigma_s^2 = \sigma_q^2 \sigma_k^2 + C_{cross}$, where $C_{cross}$ is the cross-covariance of the squared queries and keys (Goodman, 1960). Then we have*

$$\sigma_a \le \sqrt{\frac{n}{2\sqrt{1 + \frac{1}{e^{2\sigma_s^2}}}(n-1)^{\frac{3}{2}} + 1}}, \tag{3}$$

*where $\sigma_a$ decreases with $n$ increasing.*

The proof of Theorem 3 is provided in Appendix B.2. Theorem 3 builds on the work of Fenton (1960); Nahshan et al. (2024), which addresses the sum of log-normal variables. As the input length $n$ increases, $\sigma_a$ decreases, leading to the suppression of more high-frequency information and a reduction in feature expressiveness due to the low-pass filtering effect of the self-attention matrix. To validate our conclusions, we sample texts of varying lengths and plot the $\sigma_a$ values of the attention matrix from the final layer of the model after inputting, as presented in Figure 7. The results indicate that $\sigma_a$ decreases as the text length increases, ultimately leading to a higher filtering rate.

To facilitate further analysis, we define the temperature of the SA defined in Nahshan et al. (2024) as:

$$\tau_s = \frac{1}{\sigma_s} = \frac{1}{\sqrt{\sigma_q^2 \sigma_k^2 + C_{cross}}}. \tag{4}$$

Then denote $\tilde{p}_{ij} = p_{ij}/\sigma_s$ and each element in attention matrix $\boldsymbol{A}$ can be rewritten as follows:

$$\boldsymbol{A}_{ij} = \frac{e^{\tilde{p}_{ij}/\tau_s}}{\sum_{k=1}^{n} e^{\tilde{p}_{ik}/\tau_s}}, \tag{5}$$

where $\tilde{p}_{ij} \sim \mathcal{N}(0,1)$ and $\sigma_a$ increases with $\tau_s$ decreases. This implies that with a lower temperature $\tau_s$, the self-attention (SA) mechanism preserves more high-frequency components in the token signals, thereby preventing collapse in long texts.

**Corollary 4.** *(Length Collapse in Text Embeddings) Given two texts of length $n$, the cosine similarity of their text embeddings tends to increase as $n$ grows.*

The proof of Corollary 4 can be found in Appendix B.3 and is primarily based on the assumption that the mean of word embeddings in natural language texts maintains a relatively consistent representation. Specifically, as illustrated in Figure 3, we get the text embedding by averaging the word embeddings derived from the BGE model's embedding matrix for a given text. We then assess the similarity within specified length intervals. The results indicate that as text length increases, the similarity between any two embeddings also rises significantly, supporting the validity of the assumption. According to Theorem 3, with an increase in text length, text embeddings tend to converge more rapidly towards the consistent representation stated in our assumption, ultimately resulting in higher similarity for longer texts. Additionally, we analyze text embeddings for certain special sequences, such as repeated tokens. Our findings reveal that even when two texts have no overlapping tokens, the cosine similarity of their text embeddings approaches 1 as the sequence length increases. A comprehensive analysis of this phenomenon can be found in Appendix E.5. This further corroborates the existence of length collapse, even when the two sequences exhibit different mean word embeddings on average.

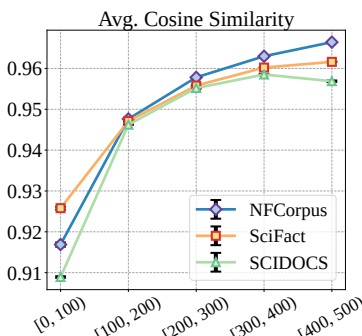

Figure 3: Mean pairwise cosine similarity of text embeddings across length intervals, with embeddings computed as the mean of token embeddings from the model's word embedding matrix.

### 2.3 DISCUSSION

**Other Components in Transformer.** After discussing how self-attention contributes to length collapse, we proceed to examine the influence of other modules in the Transformer, such as multi-head, residual, and FFN. Fortunately, previous work (Wang et al., 2022b) has talked about whether

these components can effectively alleviate the low-pass filtering drawbacks. The proof demonstrates that while these components help preserve high-frequency signals, they do not alter the fact that the MSA block, as a whole, functions solely with the representational power of a low-pass filter. Furthermore, the ability of these models to preserve high-frequency signals is solely determined by their internal parameters and architecture, independent of the input text length. As a result, these modules do not impact our analysis of length collapse in practical models.

**Difference from Over-Smoothing in Deeper Layers.** In previous research (Wang et al., 2022b), it has been noted that a self-attention module acts as a low-pass filter, causing input feature maps to gradually lose high-frequency signals as the model layers go deeper. Furthermore, other studies (Oono & Suzuki; Cai & Wang, 2020) indicate that the node features of Graph Convolutional Networks (GCNs) can become exponentially trapped in the null space of the graph Laplacian matrix. The root cause of this phenomenon is that both graph Laplacian matrices and self-attention matrices consistently exhibit a dominant eigenvector, commonly referred to as the DC component. While these studies address over-smoothing in deeper layers, we focus on how the low-pass filtering process changes as the input sequence lengthens, specifically examining over-smoothing in longer sequences.

## 3 HOW DOES LENGTH COLLAPSE LEAD TO PERFORMANCE DEGRADATION

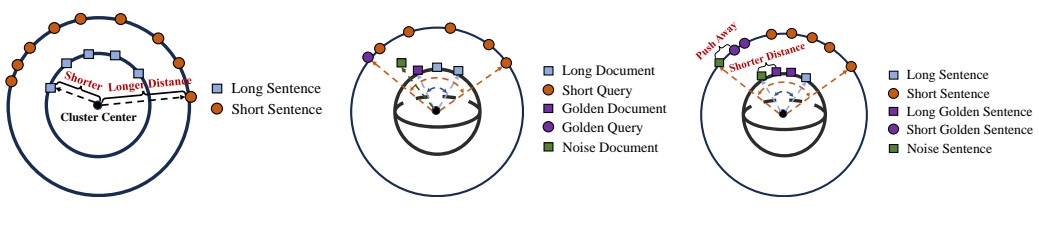

(a) Classification and Clustering        (b) Retrieval        (c) STS

Figure 4: A 3D toy example explains why length collapse leads to performance degradation.

The tasks in the MTEB benchmark can be divided into two categories: classification and clustering, and matching. For classification and clustering tasks, a classifier needs to be trained on text embeddings to categorize and cluster embeddings of different classes. Matching tasks, on the other hand, typically involve calculating the similarity between two text embeddings using cosine similarity or dot product. Furthermore, since the texts used for similarity calculations can vary greatly in length, we further divide matching-based methods into two types: retrieval and STS. In the MTEB benchmark, retrieval tasks include retrieval, longembed retrieval, and rerank, where the two texts being matched have significant length differences. The other type includes STS and summarization, where the two texts have similar lengths. Next, we will discuss these three types of tasks to analyze how the distributional differences in embeddings of long and short texts caused by length collapse impact performance across different tasks.

**Impact on classification and clustering tasks.** As shown in Figure 4 (Left), length collapse causes long-text embeddings to cluster near the center, while short-text embeddings are more dispersed around the periphery. In KNN classification, this makes clustering centers closer to long texts, resulting in the classifier being more influenced by long texts and thus reducing its performance.

**Impact on retrieval tasks.** As shown in Figure 4 (Middle), in retrieval tasks, documents are generally longer and thus distributed within a smaller, central space, while shorter queries are positioned more toward the periphery with more contextual representations (Ethayarajh, 2019). Although long documents, positioned centrally, have higher similarity with all embeddings, their representational space is also more limited. In this scenario, if there is a shorter noise document (Green Square), it may be more likely to appear relevant to the query (Purple Circle)—not because it is actually more relevant, but because its larger representational space gives it an advantage over longer documents. To verify this, we analyze the NFCorpus dataset, comparing the ranking distributions of the top 10% longest relevant documents and the top 10% shortest relevant documents, as shown in Figure 5. The results show that short documents, due to their more contextual representation, exhibit an inverted U-shaped distribution, meaning that relevant short documents are more likely to appear either at the top or the bottom of the ranking list. In contrast, long documents, with their collapsed distribution,

tend to display a more uniform distribution. Additionally, with their generally higher average cosine similarity, long documents are more likely to rank higher overall.

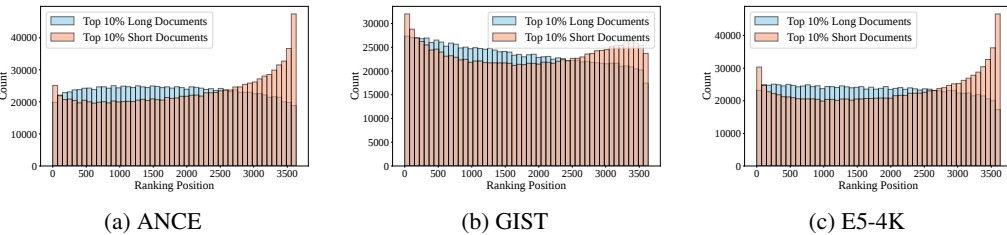

(a) ANCE          (b) GIST          (c) E5-4K

Figure 5: The count of relevant documents at different ranking positions, where a smaller ranking position indicates a document is more relevant to the query.

**Impact on STS tasks.** As shown in Figure 4 (Right), in STS tasks, length collapse causes all long-text embeddings to cluster within a narrower space. In this compact space, any two unrelated embeddings may exhibit high similarity, sometimes even surpassing that of a relevant text pair. Due to length collapse, noise sentences (Green Square) in the long-text embedding space may show higher similarity than actual related sentences (Purple Square). In contrast, in the short-text embedding space, the average distance between embeddings is larger, creating more separation between noise sentences and short sentences, while related sentences maintain high similarity due to their shared semantics. This results in lower performance for long texts compared to short texts.

# 4 MITIGATING LENGTH COLLAPSE VIA TEMPERATURE SCALING

As discussed in Section 2.2, self-attention matrix perform low-pass filtering, which narrows the filter space embedding model can express. Furthermore, distributional differences between embeddings of short texts and long texts contribute to performance degradation on longer texts. To address the problem, an intuitive idea is to increase the diversity of embeddings for long texts, making them more distinguishable within the space. Inspired by Eqn. 5, we propose a scaling technique, called Temperature Scaling (**TempScale**), which directly manipulates the attention map by multiplying $\tau_s$ by a constant temperature $\tau$ less than 1. This slows down the filtering process by increasing $\sigma_a$.

Based on Eqn. 4, a smaller $\tau_s$ results in a larger $\sigma_s$ which will further result in a larger $\sigma_a$ based on Eqn. 3. In other words, the low-pass filtering rate of the attention matrix decreases as $\tau_s$ decreases. Inspired by this, TempScale introduces a temperature $\tau$ to re-scale the self-attention matrix $\boldsymbol{A}$. Formally, let $\boldsymbol{A} = \mathrm{softmax}\left(\frac{\boldsymbol{X}\boldsymbol{W}_Q(\boldsymbol{X}\boldsymbol{W}_K)^\top}{\sqrt{d}}\right)$ denote a self-attention matrix. To decrease the low-pass filtering rate of $\boldsymbol{A}$, we apply a temperature coefficient $\tau$ to the logits before performing the softmax operation. Specifically, for each row $\boldsymbol{p}_i$ in the attention score matrix $\frac{\boldsymbol{X}\boldsymbol{W}_Q(\boldsymbol{X}\boldsymbol{W}_K)^\top}{\sqrt{d}}$, we compute the scaled logits by dividing by a temperature $\tau \in (0, 1]$, and then apply the softmax function to obtain the attention weights:

$$\boldsymbol{A} = \mathrm{softmax}\left(\frac{\boldsymbol{X}\boldsymbol{W}_Q(\boldsymbol{X}\boldsymbol{W}_K)^\top}{\tau\sqrt{d}}\right), \tag{6}$$

where a lower $\tau$ results a smaller $\tau_s$ and furthur a smaller rate of low pass filtering.

**Intuitive Explanation.** We present the effects of temperature scaling on two extreme cases to illustrate how TempScale works. As shown in Figure 6, when scaling matrix $\boldsymbol{A}$ with a relatively large $\tau$, the elements in the final matrix $\boldsymbol{A}$ can be approximated as nearly equal. In this scenario, matrix $\boldsymbol{A}$ applied to $\boldsymbol{X}$ filters out all high-frequency information, causing all token embeddings to become identical. Similarly, when scaling with a smaller temperature, matrix $\boldsymbol{A}$ no longer acts as a weighted sum of all token representations but instead selects the representation of a particular token, allowing for the retention of more high-frequency information. If we view the attention matrix as an adjacency matrix, a higher temperature leads to a denser graph, facilitating more information exchange between nodes and resulting in the loss of high-frequency information (Oono & Suzuki; Cai & Wang, 2020). In contrast, a lower temperature produces a sparser graph, allowing self-attention to preserve more high-frequency information at individual nodes and prevent over-smoothing between nodes. For further discussion on TempScale and other methods, see Appendix F.

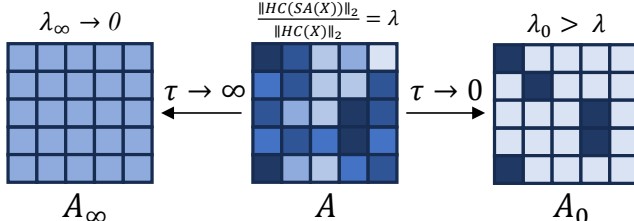
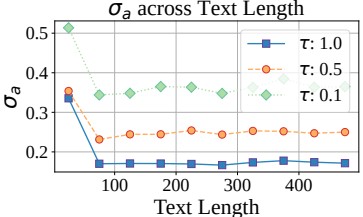

Figure 6: Two extreme cases of TempScale: larger $\tau$ causes uniform matrix elements in self-attention matrix $A$, filtering out high-frequency information, while smaller $\tau$ preserves high-frequency details by selecting specific token representations. The darker the color, the higher the attention score.

Figure 7: Visualization of the value of $\sigma_a$ across different text length. $\tau$ is defined in Eqn. 6 for scaling. See more details in Appendix E.3

## 5 EXPERIMENTS

In this section, we first conduct experiments to validate the effectiveness of our TempScale on MTEB (Muennighoff et al., 2023) and LongEmbed. Then we analyze how different tasks can benefit from TempScale to validate our theoretical analysis.

### 5.1 CAN TEMPSCALE BENEFIT EMBEDDING MODELS?

**Experiment Settings.** For long context retrieval, we use 4 real-world tasks curated from long-form QA and summarization in LongEmbed (Zhu et al., 2024) to access embedding models' factuality in short-query and long-document settings. For other embedding tasks, we consider six other tasks, including classification, clustering, summarization, semantic textual similarity (STS), retrieval, and reranking, comprising 36 datasets from MTEB (Muennighoff et al., 2023). To comprehensively evaluate TempScale, we select several representative Transformer-based embedding models, including: (1) ANCE (Xiong et al., 2021); (2) GTR (Ni et al., 2022); (3) GIST (Solatorio, 2024); (4) BGE (Xiao et al., 2023); (5) E5 (Zhu et al., 2024). These models are fine-tuned from various pretrain language models, including BERT (Kenton & Toutanova, 2019), RoBERTa (Liu et al., 2019), and T5 (Chung et al., 2024). More descriptions of the datasets and models can be found in Appendix D. When evaluating the embedding models, we set the same $\tau$ on the softmax function for the attention modules across all layers within the range of $\{0.1, 0.5, 0.6, 0.7, 0.8, 0.9, 1.0\}$. Unless otherwise specified, all temperatures used for TempScale are set to $0.8$. The metrics used for different tasks are consistent with MTEB and can be found in Appendix C.

**Results.** We select the optimal temperature $\tau$ for each model based on their performance across all tasks and organize the experimental results as shown in Table 1. The results show that these embedding models can benefit from our proposed method TempScale across various general tasks with an average improvement of 0.53% and across long context retrieval datasets with an average improvement of 0.82%. In these tasks, some datasets, such as those for STS, have an average text length of only around 10 tokens, whereas the texts in LongEmbed generally exceed 1000 tokens. Our method proves effective across both, demonstrating that using TempScale not only prevents long text collapse but also enhances the embeddings for short texts, leading to improved performance in downstream tasks. In addition, larger context window sizes lead to greater performance improvements, with E5 showing the highest improvement of 1.07%. This could be attributed to larger windows providing more and longer data for adjustment.

### 5.2 FURTHER ANALYSIS

**How do the classification and clustering tasks benefit from TempScale?** In Section 3, we attribute the performance decline in classification and clustering tasks to the distributional differences between long and short texts, which lead the model to assign greater weight to long text embeddings during classifier training. TempScale addresses this by adjusting the longer texts to align with the same space as short texts as described in Figure 1c, ensuring that both contribute equally during training. More experiments can be found in Appendix E.4.

Table 1: Average of the main metric (see Appendix C) per task on MTEB English subsets and LongEmbd. Relative Improv. means percentage increase over the performance without TempScale and improvements are highlighted with ▲ while decreasing values are denoted by ▼.

| Num. Datasets ($\rightarrow$) | Class. 8 | Clust. 11 | Summ. 1 | STS 10 | BeirRetr. 2 | Rerank. 4 | LongEmbdRetr. 4 | Avg. 40 |
|---|---|---|---|---|---|---|---|---|
| *window=512* | | | | | | | | |
| ANCE | 55.27 | 33.04 | 29.58 | 66.32 | 36.87 | 49.09 | 34.02 | 43.45 |
| +ours($\tau = 0.9$) | 55.37 | 33.28 | 29.56 | 66.47 | 36.86 | 49.25 | 33.93 | 43.53 |
| Relative Improv. (%) | 0.17 ▲ | 0.73 ▲ | -0.05 ▼ | 0.22 ▲ | -0.01 ▼ | 0.32 ▲ | -0.25 ▼ | 0.18 ▲ |
| GTR | 55.10 | 38.65 | 29.67 | 70.11 | 44.98 | 54.23 | 37.33 | 47.15 |
| +ours($\tau = 0.8$) | 55.51 | 39.52 | 29.83 | 70.26 | 45.61 | 54.16 | 37.33 | 47.46 |
| Relative Improv. (%) | 0.73 ▲ | 2.26 ▲ | 0.54 ▲ | 0.21 ▲ | 1.41 ▲ | -0.13 ▼ | 0.01 ▲ | 0.65 ▲ |
| GIST | 64.75 | 44.77 | 31.14 | 75.61 | 52.77 | 58.55 | 38.21 | 52.26 |
| +ours($\tau = 0.9$) | 65.00 | 44.64 | 31.17 | 75.59 | 53.41 | 58.60 | 38.35 | 52.39 |
| Relative Improv. (%) | 0.38 ▲ | -0.29 ▼ | 0.09 ▲ | -0.03 ▼ | 1.21 ▲ | 0.08 ▲ | 0.36 ▲ | 0.26 ▲ |
| BGE | 64.79 | 45.80 | 31.03 | 75.88 | 55.29 | 58.87 | 37.46 | 52.73 |
| +ours($\tau = 0.8$) | 64.89 | 45.61 | 31.51 | 75.68 | 56.00 | 58.97 | 38.35 | 53.00 |
| Relative Improv. (%) | 0.16 ▲ | -0.42 ▼ | 1.53 ▲ | -0.26 ▼ | 1.29 ▲ | 0.17 ▲ | 2.40 ▲ | 0.51 ▲ |
| *window=4k* | | | | | | | | |
| E5 | 61.72 | 38.82 | 30.58 | 71.77 | 47.22 | 53.12 | 56.01 | 51.32 |
| +ours($\tau = 0.8$) | 62.15 | 40.22 | 31.11 | 72.17 | 47.06 | 53.47 | 56.88 | 51.87 |
| Relative Improv. (%) | 0.70 ▲ | 3.61 ▲ | 1.74 ▲ | 0.55 ▲ | -0.33 ▼ | 0.65 ▲ | 1.56 ▲ | 1.07 ▲ |
| Avg Improv. (%) | 0.43 ▲ | 1.18 ▲ | 0.77 ▲ | 0.14 ▲ | 0.71 ▲ | 0.22 ▲ | 0.82 ▲ | 0.53 ▲ |

**How do the retrieval tasks benefit from TempScale?** In Section 3, we attribute the performance decline in retrieval tasks to the fact that short texts have more contextual embeddings. Therefore, TempScale improves the performance of long texts by adjusting the temperature to increase high-frequency information, enabling a more contextual distribution. To validate this, we select NFCorpus and SciFact to investigate the impact. As shown in Table 2, we record the average ranking position of the longest positive documents after applying TempScale at different temperatures, where a lower value indicates the documents are ranked higher. The experimental results show that a lower temperature causes the model to rank relevant long documents higher. Our method enables long texts to become more contextual, thereby reducing the bias introduced by length. In summary, TempScale enhances the performance of long documents by introducing TempScale. Similar phenomena on more models and datasets are provided in Appendix E.6.

Table 2: Average ranking position with 20% longest document across different temperature $\tau$.

| (a) NFCorpus | | | | | (b) SciFact | | | |
|---|---|---|---|---|---|---|---|---|
| **Temperature $\tau$** | 1.0 | 0.9 | 0.8 | | **Temperature $\tau$** | 1.0 | 0.9 | 0.8 |
| ANCE | 1,306.2 | 1,291.6 | 1,278.1 | | ANCE | 80.7 | 78.7 | 81.5 |
| GTR | 1,132.8 | 1,120.5 | 1,113.1 | | GTR | 81.7 | 76.6 | 72.4 |
| GIST | 1,111.0 | 1,112.8 | 1,113.8 | | GIST | 14.4 | 12.4 | 11.3 |
| BGE | 998.3 | 978.9 | 965.1 | | BGE | 13.3 | 12.3 | 12.4 |
| E5 | 1,193.3 | 1,172.3 | 1,162.8 | | E5 | 66.8 | 47.5 | 38.9 |

**How do the STS tasks benefit from TempScale?** TempScale enhances performance by giving long-document embeddings the same spatial representation as short texts. To verify this, we record the cosine similarity between random sentences, related sentences, and unrelated sentences at different temperatures, as shown in Table 3. The results show that as the temperature in TempScale decreases, the similarity between related sentences remains relatively high, while the similarity between unrelated sentences is further reduced. Specifically, although the similarity between random sentences increases with temperature in STS13 and STS14, the increase in the unrelated part is not as large as that of the random sentences. This indicates that the distance between unrelated sentences is increasing.

**Can the temperature be set based on the length of the text?** As shown in Table 2, the ranking of relevant long documents in retrieval tasks improves as the temperature decreases. This indicates that using a smaller temperature $\tau$ for long texts can better align the embeddings of long and short texts within the same distribution. For longer texts, better performance can be achieved by setting a smaller $\tau$, and the discussions can be found in Appendix E.6.

Table 3: Average cosine similarities across different temperature settings.

| Temperature | STS12 | | | STS13 | | | STS14 | | |
|---|---|---|---|---|---|---|---|---|---|
| | Random | Related | Unrelated | Random | Related | Unrelated | Random | Related | Unrelated |
| 1.0 | 0.9097 | 0.9611 | 0.7973 | 0.8707 | 0.9470 | 0.7931 | 0.8877 | 0.9494 | 0.7889 |
| 0.9 | 0.9096 | 0.9612 | 0.7937 | 0.8721 | 0.9481 | 0.7942 | 0.8886 | 0.9499 | 0.7893 |
| 0.8 | 0.9097 | 0.9612 | 0.7926 | 0.8741 | 0.9490 | 0.7969 | 0.8897 | 0.9501 | 0.7912 |

## 6 RELATED WORK

**Text Embedding Models.** Embeddings are generated by pre-trained language models (PLMs), which produce fixed-dimensional embeddings regardless of the input text length, laying the foundation of numerous NLP applications. Early works on text embeddings lack context awareness and are thus commonly labeled as word embedding models (Pennington et al., 2014). Modern embedding models (Wang et al., 2022a; Xiao et al., 2023) incorporate context awareness into language models through self-attention mechanisms, serving as the foundation for the latest embedding models. These models, based on Transformer architectures like BERT (Kenton & Toutanova, 2019) and RoBERTa (Liu et al., 2019), are first pre-trained on large-scale weakly supervised text pairs using contrastive loss (Gao et al., 2021), then fine-tuned on small scale but high-quality datasets. More recently, (Muennighoff et al., 2024) investigates the integration of generative and embedding tasks in large language models, introducing transformer-based GritLM, which enhances performance in both areas. Among the diverse range of pre-trained transformers (Wolf et al., 2020), self-attention plays a crucial role, and this paper focuses on how this module contributes to length collapse.

**Context Window Extension for Embedding Model.** Despite transformer-based embedding models excelling in generating vector representations, they are typically constrained by a narrow context window of around 512 input tokens (Wang et al., 2022a; Xiao et al., 2023; Ni et al., 2022). This limitation significantly restricts their use in scenarios that require processing long inputs, such as extensive Wikipedia entries or meeting transcripts (Saad-Falcon et al., 2024; Zhu et al., 2024). Previous work attributes this poor performance to limited context windows and attempts to extend the window size. Current initiatives to develop long-context embedding models generally begin with acquiring a long-context backbone model, either by pre-training from scratch with long inputs (Günther et al., 2023; Nussbaum et al., 2024; Chen et al., 2024) or utilizing existing models (Wang et al., 2024; Zhu et al., 2024). This is followed by training the backbone model to generate embeddings. However, in this paper, we discover that regardless of the model's context window size, the model consistently performs worse on longer texts than on shorter texts due to length collapse. Therefore, we aim to improve the performance of long texts across all context window size models by analyzing and addressing length collapse, rather than simply expanding context window size.

## 7 CONCLUSION AND FUTURE WORK

In this paper, we identify the phenomenon of length collapse, where embeddings of longer texts tend to cluster together and propose practical solutions through Fourier domain analysis. Our theoretical findings suggest that Multi-Head Self-Attention inherently performs stronger low-pass filtering as the token sequence length increases, leading to patch uniformity issues in longer sentences. To this end, we propose a technique called TempScale, which effectively reduces the low-pass filtering effect by introducing a temperature parameter when applying softmax in the self-attention matrix. Our extensive experiments validate the effectiveness of our methods and enhance the performance of general embedding models on the MTEB and LongEmbed benchmarks. Overall, TempScale is a crucial advancement in enhancing the performance of embedding models on longer texts.

Future work includes: **1) LLM-based embedding model:** Although we have observed Length Collapse in LLM-based embedding models in Appendix G, further analysis is needed to investigate how unidirectional attention mechanisms specifically contribute to this phenomenon; **2) Tuning method:** The work in this paper relies on existing models and pre-trained parameters without using a training dataset. In future work, we will focus on tuning the temperature for additional improvements. **3) Analysis on more modules:** We primarily investigate the impact of the self-attention module on length collapse in this paper. Moving forward, we plan to explore the effects of additional modules in transformers such as LayerNorm and FFN.

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

## A BACKGROUND INFORMATION ABOUT FOURIER ANALYSIS

In this appendix, we provide additional background information on Fourier analysis. Specifically, consider the discrete Fourier transform (DFT) in the real-valued domain, denoted as $\mathcal{F}: \mathbb{R}^n \to \mathbb{C}^n$. The DFT can be expressed in matrix form as shown below:

$$
\boldsymbol{DFT} = \frac{1}{\sqrt{n}} \begin{bmatrix} 1 & 1 & \cdots & 1 \\ 1 & e^{2\pi \mathrm{j}} & \cdots & e^{2\pi \mathrm{j}(n-1)} \\ \vdots & \vdots & \ddots & \vdots \\ 1 & e^{2\pi \mathrm{j}(k-1)\cdot 1} & \cdots & e^{2\pi \mathrm{j}(k-1)\cdot(n-1)} \\ \vdots & \vdots & \ddots & \vdots \\ 1 & e^{2\pi \mathrm{j}(n-1)} & \cdots & e^{2\pi \mathrm{j}(n-1)^2} \end{bmatrix},
$$

and inverse discrete Fourier transform is $\boldsymbol{DFT}^{-1} = \boldsymbol{DFT}^\top = \overline{\boldsymbol{DFT}}$. In signal processing, we can regard matrices as multi-channel signals. For example, $\boldsymbol{X} \in \mathbb{R}^{n \times d}$ means $d$-channel $n$-length signals. When the DFT and inverse DFT are applied to multi-channel signals, each channel is transformed independently. That is, $\mathcal{F}(\boldsymbol{X}) = [\mathcal{F}(\boldsymbol{x}_1) \quad \cdots \quad \mathcal{F}(\boldsymbol{x}_d)] = \boldsymbol{DFT} \cdot \boldsymbol{X}$.

Hereby, we can independently operators $\mathcal{DC}[\cdot]$ and $\mathcal{HC}[\cdot]$ on echo channel using the matrices in Eqn. 7. Then we can write $\mathcal{DC}[\cdot]$ as below:

$$
\mathcal{DC}[\boldsymbol{x}] = \boldsymbol{DFT}^{-1} \operatorname{diag}(1, 0, \cdots, 0) \boldsymbol{DFT} \boldsymbol{x}
$$
$$
= \frac{1}{n} \mathbf{1} \mathbf{1}^T \boldsymbol{x}.
$$

Conversely, we can denote $\mathcal{HC}[\cdot]$ as:

$$
\mathcal{HC}[\boldsymbol{x}] = \boldsymbol{DFT}^{-1} \operatorname{diag}(0, 1, \cdots, 1) \boldsymbol{DFT} \boldsymbol{x}
$$
$$
= \boldsymbol{DFT}^{-1} (\boldsymbol{I} - \operatorname{diag}(1, 0, \cdots, 0)) \boldsymbol{DFT} \boldsymbol{x}
$$
$$
= (\boldsymbol{I} - \frac{1}{n} \mathbf{1} \mathbf{1}^T) \boldsymbol{x}.
$$

## B DETAILED PROOFS

### B.1 PROOF OF THEOREM 2

We start our analysis by providing a lemma.

**Lemma 5.** *The following holds $\|\boldsymbol{AB}\|_F \leq \|\boldsymbol{A}\|_2 \|\boldsymbol{B}\|_F$ and $\|\boldsymbol{AB}\|_F \leq \|\boldsymbol{A}\|_F \|\boldsymbol{B}\|_2$.*

*Proof.* Denote $\boldsymbol{B} = (\boldsymbol{b_1} \quad \cdots \quad \boldsymbol{b_n})$ and we have $\boldsymbol{AB} = (\boldsymbol{Ab_1} \quad \cdots \quad \boldsymbol{Ab_n})$. From the definition of the spectral norm, we have: $\|\boldsymbol{A}\|_2 \geq \frac{\|\boldsymbol{Ab_i}\|_2}{\|\boldsymbol{b_i}\|_2}$. Taking the average of the right-hand side, we obtain: $\|\boldsymbol{A}\|_2^2 \geq \sum_{i=1}^n \frac{\|\boldsymbol{b_i}\|_2^2}{\|\boldsymbol{B}\|_F^2} \frac{\|\boldsymbol{Ab_i}\|_2^2}{\|\boldsymbol{b_i}\|_2^2}$. This implies: $\|\boldsymbol{A}\|_2^2 \|\boldsymbol{B}\|_F^2 \geq \sum_{i=1}^n \|\boldsymbol{Ab_i}\|_2^2 = \|\boldsymbol{AB}\|_F^2$. Finally, the last step utilizes the result $\|\boldsymbol{A}\|_F^2 = \sum_{i,j} |a_{ij}|^2 = \sum_{j=1}^n \|\boldsymbol{a_j}\|_2^2$. Because both the spectral norm and the Frobenius norm of a matrix remain unchanged under transposition, we have $\|\boldsymbol{AB}\|_F = \|\boldsymbol{B}^\top \boldsymbol{A}^\top\|_F \leq \|\boldsymbol{B^T}\|_2 \|\boldsymbol{A^T}\|_F = \|\boldsymbol{A}\|_F \|\boldsymbol{B}\|_2$.

$\square$

**Theorem 6.** *(Filter Rate of SA) Let $\sigma_a$ be the largest singular value of $\mathcal{HC}\left[\boldsymbol{A}\right]$. Define* $\mathrm{SA}(\boldsymbol{X}) = \boldsymbol{AXW}_V$ *as the output of a self-attention module, then*

$$\|\mathcal{HC}\left[\mathrm{SA}(\boldsymbol{X})\right]\|_F \leq \sigma_a \|\boldsymbol{W}_V\|_2 \|\mathcal{HC}\left[\boldsymbol{X}\right]\|_F. \tag{7}$$

*Proof.* First, we write $\boldsymbol{X} = \mathcal{DC}\left[\boldsymbol{X}\right] + \mathcal{HC}\left[\boldsymbol{X}\right] = \frac{1}{n}\boldsymbol{11}^\top \boldsymbol{X} + \boldsymbol{H}$, where $\boldsymbol{H} = \mathcal{HC}\left[\boldsymbol{X}\right]$ represents the remaining part of the original signals.

$$\mathcal{HC}\left[\mathrm{SA}(\boldsymbol{X})\right] = \left(\boldsymbol{I} - \frac{1}{n}\boldsymbol{11}^T\right)\boldsymbol{AXW}_V \tag{8}$$

$$= \left(\boldsymbol{I} - \frac{1}{n}\boldsymbol{11}^T\right)\boldsymbol{A}(\frac{1}{n}\boldsymbol{11}^\top \boldsymbol{X} + \boldsymbol{H})\boldsymbol{W}_V \tag{9}$$

$$= \frac{1}{n}\left(\boldsymbol{I} - \frac{1}{n}\boldsymbol{11}^T\right)\boldsymbol{A11}^\top \boldsymbol{XW}_V + \left(\boldsymbol{I} - \frac{1}{n}\boldsymbol{11}^T\right)\boldsymbol{AHW}_V \tag{10}$$

$$= \left(\boldsymbol{I} - \frac{1}{n}\boldsymbol{11}^T\right)\boldsymbol{AHW}_V \tag{11}$$

Therefore,

$$\|\mathcal{HC}\left[\mathrm{SA}(\boldsymbol{X})\right]\|_F = \left\|\left(\boldsymbol{I} - \frac{1}{n}\boldsymbol{11}^T\right)\boldsymbol{AHW}_V\right\|_F \tag{12}$$

$$\leq \left\|\left(\boldsymbol{I} - \frac{1}{n}\boldsymbol{11}^T\right)\boldsymbol{A}\right\|_2 \|\boldsymbol{W}_V\|_2 \|\boldsymbol{H}\|_F \tag{13}$$

$$= \sigma_a \|\boldsymbol{W}_V\|_2 \|\boldsymbol{H}\|_F \tag{14}$$

The Eqn. 13 leverages inequality in Lemma 5.

$\square$

## B.2 PROOF OF THEOREM 3

**Theorem 7.** *(Filter Rate of Different Input Length $n$) Let $\boldsymbol{XW}_Q$ and $\boldsymbol{XW}_K$ be a Gaussian matrix, where elements $q_{ij} \sim \mathcal{N}(0, \sigma_q^2)$ and $k_{ij} \sim \mathcal{N}(0, \sigma_k^2), \forall i, j$. Let $x_{ij} = \boldsymbol{q}_i^\top \boldsymbol{k}_j / \sqrt{d}$ the attention score of pair $i, j$, whose variance can be expressed as $\sigma_s^2 = \sigma_q^2 \sigma_k^2 + C_{cross}$, where $C_{cross}$ is the cross-covariance of the squared queries and keys (Goodman, 1960). Then we have*

$$\sigma_a \leq \sqrt{\frac{n}{2\sqrt{1 + \frac{1}{e^{2\sigma_s^2}}}(n-1)^{\frac{3}{2}} + 1}}, \tag{15}$$

*where $\sigma_a$ decreases with $n$ increasing.*

*Proof.* First we have

$$\sigma_a = \left\| \left( \boldsymbol{I} - \frac{1}{n}\mathbf{1}\mathbf{1}^T \right) \boldsymbol{A} \right\|_2 \tag{16}$$

$$\leq \left\| \boldsymbol{I} - \frac{1}{n}\mathbf{1}\mathbf{1}^T \right\|_2 \|\boldsymbol{A}\|_2 \tag{17}$$

$$\leq \|\boldsymbol{A}\|_F \tag{18}$$

The Eqn. 18 leverages $\left\| \boldsymbol{I} - \frac{1}{n}\mathbf{1}\mathbf{1}^T \right\|_2 = 1$ and $\|\boldsymbol{A}\|_2 \leq \|\boldsymbol{A}\|_F$. Now we need to upper bound $\|\boldsymbol{A}\|_F$. Generally, the product of two independent Gaussian variables has a density in the form of a modiffed Bessel function of the second kind (Nahshan et al., 2024). When the vector dimensions are sufficiently large, the Central Limit Theorem implies that the distribution of the dot product between $\boldsymbol{q}_i$ and $\boldsymbol{k}_j$ can be approximated by a Gaussian distribution with zero mean and variance $\sigma_s^2$. As mentioned in Theorem 3, the variance of $\boldsymbol{q}_i^\top \boldsymbol{k}_j$ can be expressed as $\sigma^2 = \sigma_q^2 \sigma_k^2 + C_{\text{cross}}$, where $C_{\text{cross}} = \text{Cov}(\boldsymbol{q}^2, \boldsymbol{k}^2) - \text{Cov}(\boldsymbol{q}, \boldsymbol{k})^2$ is the cross-covariance of the squared queries and keys (Goodman, 1960). Thus we can suppose that each element $p_j \sim \mathcal{N}(0, \sigma_s)$ in the matrix $\boldsymbol{X}\boldsymbol{W_Q}\boldsymbol{W_K^\top}\boldsymbol{X^T}$ is independent, where $j \in (1, \cdots, n)$:

$$\|\boldsymbol{A}\|_F = \sqrt{n \sum_{j=1}^n \left( \frac{e^{x_j}}{\sum_{i=1}^n e^{x_i}} \right)^2} \tag{19}$$

$$= \sqrt{n \frac{\sum_{j=1}^n e^{2x_j}}{2\sum_{i=1}^n \sum_{j=1, j\neq i}^n e^{x_i + x_j} + \sum_{j=1}^n e^{2x_j}}} \tag{20}$$

$$= \sqrt{\frac{n e^{\ln n + 2\sigma_s^2 - \frac{1}{2}\ln\frac{e^{4\sigma_s^2}-1}{n}}}{2e^{\ln n(n-1) + \sigma_s^2 - \frac{1}{2}\ln\frac{e^{2\sigma_s^2}-1}{n(n-1)}} + e^{\ln n + 2\sigma_s^2 - \frac{1}{2}\ln\frac{e^{4\sigma_s^2}-1}{n}}}} \tag{21}$$

$$= \sqrt{\frac{n}{2\sqrt{1 + \frac{1}{e^{2\sigma_s^2}}}(n-1)^{\frac{3}{2}} + 1}}. \tag{22}$$

The derivation in Eqn. 21 primarily relies on the theorem from Fenton (1960), which addresses the sum of log-normal variables. First, we have $e^x \sim \text{LogNormal}(0, \sigma_s^2)$ and $e^{2x} \sim \text{LogNormal}(0, 4\sigma_s^2)$. Additionally, we assume that $e^{x_i}$ and $e^{x_j}$ are independent, leading to $e^{x_i + x_j} \sim \text{LogNormal}(0, 2\sigma_s^2)$. Now, considering the sum of log-normal variables, Fenton (1960)'s theorem provides that, for moderate values of $\sigma^2$, the sum of zero-mean i.i.d. log-normal variables can be approximated by another log-normal distribution with mean $\mu_\Sigma$ and variance $\sigma_\Sigma^2$, where:

$$\sigma_\Sigma^2 = \ln\left( \frac{1}{n}\left( e^{\sigma^2} - 1 \right) + 1 \right); \quad \mu_\Sigma = \ln n + (\sigma^2 - \sigma_\Sigma^2)/2.$$

For moderate values of $n$ and $\sigma^2$, the variance $\sigma_\Sigma^2$ can be approximated as:

$$\sigma_\Sigma^2 \approx \ln\left( \frac{1}{n}\left( e^{\sigma^2} - 1 \right) \right).$$

Thus, the sum $\sum_{j=1}^n e^{2x_j}$ follows a log-normal distribution:

$$\sum_{j=1}^n e^{2x_j} \sim \text{LogNormal}\left( \ln n + 2\sigma_s^2 - \frac{1}{2}\ln\left( \frac{1}{n}\left( e^{4\sigma^2} - 1 \right) \right), \ln\left( \frac{1}{n}\left( e^{4\sigma^2} - 1 \right) \right) \right).$$

Similarly, the sum $\sum_{i=1}^n \sum_{j=1, j\neq i}^n e^{x_i + x_j}$ follows:

$$\sum_{i=1}^{n} \sum_{j=1, j \neq i}^{n} e^{x_i + x_j} \sim \text{LogNormal}\left(\ln n(n-1) + \sigma_s^2 - \frac{1}{2}\ln\left(\frac{e^{2\sigma_s^2} - 1}{n(n-1)}\right), \ln\left(\frac{e^{2\sigma_s^2} - 1}{n(n-1)}\right)\right).$$

From these, Eqn. 21 follows naturally. Finally, we have

$$\sigma_a \leq \sqrt{\frac{n}{2\sqrt{1 + \frac{1}{e^{2\sigma_s^2}}}(n-1)^{\frac{3}{2}} + 1}},$$

where $\sigma_a$ decreases with $n$ increasing.

$\square$

### B.3 PROOF OF COROLLARY 4

**Theorem 8.** *(Length Collapse in Text Embeddings) Given two texts of length $n$, the cosine similarity of their text embeddings tends to increase as $n$ grows.*

*Proof.* Given the two texts embeddings $\boldsymbol{x}_1$ and $\boldsymbol{x}_2$, we have

$$cos(\boldsymbol{x}_1, \boldsymbol{x}_2) = \frac{\left(\mathcal{HC}\left[\boldsymbol{x}_1\right] + \mathcal{DC}\left[\boldsymbol{x}_1\right]\right)\left(\mathcal{HC}\left[\boldsymbol{x}_2^T\right] + \mathcal{DC}\left[\boldsymbol{x}_2^T\right]\right)}{\|\boldsymbol{x}_1\|_2 \|\boldsymbol{x}_2\|_2} \tag{23}$$

$$= \frac{\mathcal{HC}\left[\boldsymbol{x}_1\right]\mathcal{HC}\left[\boldsymbol{x}_2^T\right] + \mathcal{DC}\left[\boldsymbol{x}_1\right]\mathcal{DC}\left[\boldsymbol{x}_2^T\right]}{\|\boldsymbol{x}_1\|_2 \|\boldsymbol{x}_2\|_2} \tag{24}$$

$$\geq \frac{\alpha^2}{\sqrt{\alpha^2 + \alpha_1^2}\sqrt{\alpha^2 + \alpha_2^2}}, \tag{25}$$

$\square$

where $\alpha_1$ and $\alpha_2$ represent the maximum values in the frequency domain of $\mathcal{HC}\left[\boldsymbol{x}_1\right]$ and $\mathcal{HC}\left[\boldsymbol{x}_2\right]$ and $\alpha$ represent the value of $\mathcal{DC}\left[\boldsymbol{x}_1\right]$ and $\mathcal{DC}\left[\boldsymbol{x}_2\right]$, respectively, after applying the discrete Fourier transform. Eqn. 24 leverages that $\mathcal{HC}\left[\cdot\right]$ and $\mathcal{DC}\left[\cdot\right]$ are orthogonal. Eqn. 25 leverages that the assumption the mean of word embeddings in natural language texts maintains a relatively consistent representation. Thus $\mathcal{HC}\left[\boldsymbol{x}_1\right]$ and $\mathcal{HC}\left[\boldsymbol{x}_2\right]$ have the same value $\alpha$ after applying the discrete Fourier transform. Finally, according to Theorem 3, $\alpha_1$ and $\alpha_2$ will gradually decrease with $n$ grows, leading to a higher cosine similarity between $\boldsymbol{x}_1$ and $\boldsymbol{x}_2$.

## C DATASETS AND EVALUATION METRICS

Table 4 provides an overview of the datasets used in our experiments. Next, we give a brief description of the tasks involved in the experiments and the corresponding datasets and evaluation metrics they include.

### C.1 CLASSIFICATION

In general, we use the provided embedding model to obtain a training set and a test set. The embeddings of the training set are used to train a logistic regression classifier with a maximum of 100 iterations, which is then scored on the test set. The main evaluation metrics are accuracy, average precision, and the f1 score.

**AmazonPolarity** (Zhang et al., 2015) consists of Amazon customer reviews, each labeled as either "positive" or "negative."

**Banking77** (Casanueva et al., 2020) dataset consists of online banking user queries labeled with one of 77 specific intents.

**Emotion** (Saravia et al., 2018) comprises Twitter messages categorized by six fundamental emotions: anger, fear, joy, love, sadness, and surprise.

**Imdb** (Maas et al., 2011) consists of extensive movie reviews categorized as either positive or negative.

**MassiveIntent** (FitzGerald et al., 2022) is a multilingual dataset featuring a diverse array of utterances from Amazon Alexa, each labeled with one of 60 different intents across 51 languages.

**MassiveScenario** (FitzGerald et al., 2022) dataset comprises a diverse collection of Amazon Alexa user utterances, each labeled with one of 60 thematic intents, and supports 51 languages.

**ToxicConversations** [1] , sourced from a Kaggle competition, comprises comments from the Civil Comments platform, complete with annotations indicating whether each comment is toxic.

**TweetSentimentExtraction** [2], a dataset from a Kaggle competition focuses on classifying tweets into three categories: neutral, positive, and negative sentiments.

## C.2 CLUSTERING

Clustering aims at grouping a given set of sentences or textbfs into meaningful clusters by training a mini-batch k-means model on the text embeddings. The model is scored using the v-measure (Rosenberg & Hirschberg, 2007). Since the v-measure does not depend on the cluster labels, the arrangement of the labels will not affect the score.

**ArxivClusteringS2S, ArxivClusteringP2P, BiorxivClusteringS2S, BiorxivClusteringP2P, MedrxivClusteringP2P, MedrxivClusteringS2S** (Muennighoff et al., 2023). These datasets are tailored for MTEB, utilizing titles or a combination of titles and abstracts from arXiv, bioRxiv, and medRxiv, with clustering labels derived from human-assigned categories, emphasizing both main and secondary classification levels.

**RedditClustering** (Geigle et al., 2021), a dataset consists of titles from 199 subreddits, is organized into 25 splits, each featuring 10 to 50 classes, with every class containing between 100 and 1,000 sentences.

**RedditClusteringP2P** (Muennighoff et al., 2023), developed for the MTEB, consists of Reddit posts combined with their titles, organized into ten splits featuring 10 and 100 clusters each, with a total of 1,000 to 100,000 posts, aimed at clustering based on subreddit affiliation.

**StackExchangeClustering** (Geigle et al., 2021), a dataset consisting of titles from 121 Stack Exchange communities, is organized into 25 subsets, each containing 10 to 50 categories, with 100 to 1,000 sentences per category.

**StackExchangeClusteringP2P** (Muennighoff et al., 2023), designed for MTEB, comprises 10 splits of posts from StackExchange, each containing 5,000 to 10,000 entries, clustered by subreddit based on the combined content of titles and posts.

**TwentyNewsgroupsClustering** [3] consists of article titles from 20 different newsgroups, designed for clustering tasks, and includes 10 splits with each split featuring between 1,000 and 10,000 titles across the 20 categories.

## C.3 RERANKING

Reranking involves inputting a query along with a series of relevant and irrelevant reference texts, then sorting the results based on their relevance to the query. The provided model embeds the reference texts, which are compared to the query using cosine similarity. Each query is scored, and the average score across all queries is used to generate the final ranking. The evaluation metrics are MRR@k and MAP, with MAP serving as the primary metric.

---

[1]ToxicConversations

[2]TweetSentimentExtraction

[3]https://scikit-learn.org/0.19/datasets/twenty_newsgroups.html

**AskUbuntuDupQuestions** [4] dataset comprises questions sourced from AskUbuntu, accompanied by manually annotated labels that indicate whether pairs of questions are similar or dissimilar.

**MindSmall** (Wu et al., 2020) dataset is a comprehensive English resource designed for research in news recommendation, focusing on ranking news articles based on the title of a currently read article to suggest related content.

**SciDocsRR** (Cohan et al., 2020) is a dataset designed for ranking related scientific papers using their titles as the primary basis for assessment.

**StackOverflowDupQuestions** (Liu et al., 2018) dataset focuses on identifying whether questions tagged with Java, JavaScript, and Python on Stack Overflow are duplicates of existing queries.

## C.4 RETRIEVAL

In retrieval task, each dataset consists of a corpus, queries, and a mapping of each query to relevant documents. The task goal is to find these relevant documents based on a given query. When evaluating, we first use the provided model to embed queries and corpus documents and then calculate the cosine similarity to obtain relevance scores and rank the corpus documents for each query based on these scores. The evaluation metrics consists of nDCG@k, MRR@k, MAP@k, precision@k, and recall@k, with nDCG@10 as the primary metric.

### C.4.1 BEIR RETRIEVAL

**NFCorpus** (Boteva et al., 2016) is a dataset that includes natural language queries sourced from NutritionFacts, paired with annotated medical documents from PubMed, utilizing the original splits from various types of content from NF, such as videos, blogs, and Q&A posts.

**SciFact** (Wadden et al., 2020) dataset evaluates scientific claims by matching them with evidence sourced from research literature, specifically utilizing a set of 300 test queries and the complete document collection from the original dataset.

### C.4.2 LONGEMBD RETRIEVAL

LongEmbed (Zhu et al., 2024) includes 4 real-world retrieval tasks curated from long-form QA and summarization. The document in LongEmbed is much longer compared to BEIR. Thus, it can effectively evaluate the capability of the embedding model on long texts.

**LEMBNarrativeQARetrieval** (Kočiskỳ et al., 2018) is a question-answering dataset featuring lengthy narratives, averaging 50,474 words, that challenge models to comprehend and extract information about characters and events dispersed throughout the stories.

**LEMBQMSumRetrieval** (Zhong et al., 2021) dataset focuses on generating summaries of meetings based on specific queries, necessitating the extraction and synthesis of relevant information from various segments of the conversation that cover multiple topics and participants.

**LEMBSummScreenFDRetrieval** (Chen et al., 2021) dataset consists of pairs of transcripts from TV series and their corresponding human-crafted summaries, requiring the integration of dispersed plot elements into concise narrative descriptions.

**LEMBWikimQARetrieval** (Ho et al., 2020) dataset is a complex question-answering resource that includes questions requiring up to five reasoning steps, designed using specific templates to encourage deep understanding rather than simple retrieval of information.

## C.5 SEMANTIC TEXTUAL SIMILARITY (STS)

The task goal is to determine the similarity between a pair of sentences, where continuous scores serve as labels, with higher values indicating greater similarity. The provided model embeds the sentences, and their similarity is calculated using cosine similarity. The primary evaluation metric is the Spearman correlation (Reimers et al., 2016).

---

[4] https://github.com/taolei87/askubuntu

Table 4: Statistics of the experimental datasets used in the work.

| Type | Name | Categ. | #Lang. | Train Samples | Dev Samples | Test Samples | Train avg. chars | Dev avg. chars | Test avg. chars |
|---|---|---|---|---|---|---|---|---|---|
| BEIR Retrieval | NFCorpus | s2p | 1 | 0 | 0 | 3,956 | 0 | 0 | 1,462.7 |
| | SciFact | s2p | 1 | 0 | 0 | 5,483 | 0 | 0 | 1,422.3 |
| Classification | AmazonPolarityClassification | p2p | 1 | 3,600,000 | 0 | 400,000 | 431.6 | 0 | 431.4 |
| | Banking77Classification | s2s | 1 | 10,003 | 0 | 3,080 | 59.5 | 0 | 54.2 |
| | EmotionClassification | s2s | 1 | 16,000 | 2,000 | 2,000 | 96.8 | 95.3 | 96.6 |
| | ImdbClassification | p2p | 1 | 25,000 | 0 | 25,000 | 1,325.1 | 0 | 1,293.8 |
| | MassiveIntentClassification | s2s | 51 | 11,514 | 2,033 | 2,974 | 35.0 | 34.8 | 34.6 |
| | MassiveScenarioClassification | s2s | 51 | 11,514 | 2,033 | 2,974 | 35.0 | 34.8 | 34.6 |
| | ToxicConversationsClassification | s2s | 1 | 50,000 | 0 | 50,000 | 298.8 | 0 | 296.6 |
| | TweetSentimentExtractionClassification | s2s | 1 | 27,481 | 0 | 3,534 | 68.3 | 0 | 67.8 |
| Clustering | ArxivClusteringP2P | p2p | 1 | 0 | 0 | 732,723 | 0 | 0 | 1,009.9 |
| | ArxivClusteringS2S | s2s | 1 | 0 | 0 | 732,723 | 0 | 0 | 74.0 |
| | BiorxivClusteringP2P | p2p | 1 | 0 | 0 | 75,000 | 0 | 0 | 1,666.2 |
| | BiorxivClusteringS2S | s2s | 1 | 0 | 0 | 75,000 | 0 | 0 | 101.6 |
| | MedrxivClusteringP2P | p2p | 1 | 0 | 0 | 37,500 | 0 | 0 | 1,981.2 |
| | MedrxivClusteringS2S | s2s | 1 | 0 | 0 | 37,500 | 0 | 0 | 114.7 |
| | RedditClustering | s2s | 1 | 0 | 420,464 | 420,464 | 0 | 64.7 | 64.7 |
| | RedditClusteringP2P | p2p | 1 | 0 | 0 | 459,399 | 0 | 0 | 727.7 |
| | StackExchangeClustering | s2s | 1 | 0 | 417,060 | 373,850 | 0 | 56.8 | 57.0 |
| | StackExchangeClusteringP2P | p2p | 1 | 0 | 0 | 75,000 | 0 | 0 | 1,090.7 |
| | TwentyNewsgroupsClustering | s2s | 1 | 0 | 0 | 59,545 | 0 | 0 | 32.0 |
| LongEmbd Retrieval | LEMBNarrativeQARetrieval | s2p | 1 | 0 | 0 | 10,804 | 0 | 0 | 326,753.5 |
| | LEMBQMSumRetrieval | s2p | 1 | 0 | 0 | 1,724 | 0 | 0 | 53,335.8 |
| | LEMBSummScreenFDRetrieval | s2p | 1 | 0 | 672 | 0 | 0 | 30,854.3 | 0 |
| | LEMBWikimQARetrieval | s2p | 1 | 0 | 0 | 500 | 0 | 0 | 37,445.6 |
| Reranking | AskUbuntuDupQuestions | s2s | 1 | 0 | 0 | 2,255 | 0 | 0 | 52.5 |
| | MindSmallReranking | s2s | 1 | 231,530 | 0 | 107,968 | 69.0 | 0 | 70.9 |
| | SciDocsRR | s2s | 1 | 0 | 19,594 | 19,599 | 0 | 69.4 | 69.0 |
| | StackOverflowDupQuestions | s2s | 1 | 23,018 | 3,467 | 3,467 | 49.6 | 49.8 | 49.8 |
| STS | BIOSSES | s2s | 1 | 200 | 200 | 200 | 156.6 | 156.6 | 156.6 |
| | SICK-R | s2s | 1 | 19,854 | 19,854 | 19,854 | 46.1 | 46.1 | 46.1 |
| | STS12 | s2s | 1 | 4,468 | 0 | 6,216 | 100.7 | 0 | 64.7 |
| | STS13 | s2s | 1 | 0 | 0 | 3,000 | 0 | 0 | 54.0 |
| | STS14 | s2s | 1 | 0 | 0 | 7,500 | 0 | 0 | 54.3 |
| | STS15 | s2s | 1 | 0 | 0 | 6,000 | 0 | 0 | 57.7 |
| | STS16 | s2s | 1 | 0 | 0 | 2,372 | 0 | 0 | 65.3 |
| | STS17 | s2s | 11 | 0 | 0 | 500 | 0 | 0 | 43.3 |
| | STS22 | p2p | 18 | 0 | 0 | 8,060 | 0 | 0 | 1,992.8 |
| | STSBenchmark | s2s | 1 | 11,498 | 3,000 | 2,758 | 57.6 | 64.0 | 53.6 |
| Summarization | SummEval | p2p | 1 | 0 | 0 | 2,800 | 0 | 0 | 359.8 |

**STS12, STS13, STS14, STS15, STS16, STS17, STS22, STSBenchmark** (Agirre et al., 2012; 2013; Bandhakavi et al., 2014; Biçici, 2015; Nakov et al., 2016) [5] [6] [7] are collections of sentence pairs designed to evaluate semantic textual similarity, with the former set focused on monolingual English pairs and the latter two incorporating cross-lingual comparisons across multiple languages.

**BIOSSES** (Soğancıoğlu et al., 2017) comprises 100 pairs of sentences specifically focused on the biomedical domain.

**SICK-R** (Dadas et al., 2020) , which stands for Sentences Involving Compositional Knowledge, comprises 100,000 diverse sentence pairs that exhibit rich lexical, syntactic, and semantic characteristics.

## C.6    SUMMARIZATION

The input consists of a set of summaries written by humans and machines. The goal is to score the machine-generated summaries. Use the provided model to embed the summaries. Calculate the distance between each machine summary and all human summary embeddings. Retain similar scores as the model score for each individual machine-generated summary. Calculate the Spearman correlation based on cosine similarity (Reimers et al., 2016) as the main metric.

**SummEval** (Fabbri et al., 2021) consists of summaries produced by advanced summarization models trained on CNN and DailyMail articles.

---

[5] https://alt.qcri.org/semeval2017/task1/
[6] https://competitions.codalab.org/competitions/33835
[7] https://github.com/PhilipMay/stsb-multi-mt/

Table 5: Embedding models used in the experiments.

| Model Name | Publicly Available Link |
| --- | --- |
| ANCE | https://huggingface.co/sentence-transformers/msmarco-roberta-base-ance-firstp |
| GTR | https://huggingface.co/sentence-transformers/gtr-t5-base |
| GIST | https://huggingface.co/avsolatorio/GIST-small-Embedding-v0 |
| BGE | https://huggingface.co/BAAI/bge-base-en-v1.5 |
| E5 | https://huggingface.co/dwzhu/e5-base-4k |

## D    EMBEDDING MODELS

Table 5 provides the models used in the experiments and their publicly available links. Below is a brief introduction to these models.

**ANCE** (Xiong et al., 2021) enhances dense retrieval by selecting challenging negative samples from the entire corpus and asynchronously updating the Approximate Nearest Neighbor (ANN) index with each training iteration, using a context window size of 512.

**GTR** (Ni et al., 2022) improves dual encoder performance for retrieval tasks by scaling up model size while keeping a fixed bottleneck embedding, leading to significant improvements in out-of-domain generalization, all within a context window size of 512.

**GIST** (Solatorio, 2024) consistently improves performance across different model sizes by leveraging the strengths of large, resource-intensive models to enhance smaller ones, making advanced AI technologies more accessible and cost-effective, all within a context window size of 512.

**BGE** (Xiao et al., 2023) offers a range of well-trained embedding models based on a BERT-like architecture, enabling users to balance performance and efficiency for various applications while also allowing easy fine-tuning. In our experiments, we use the gte-base-en-v1.5 model, which operates with a context window size of 512.

**E5** (Zhu et al., 2024) is a long-context embedding model fine-tuned to support 4k token inputs while maintaining the original performance for shorter contexts, designed to advance research in long-context embedding technologies. It uses a context window size of 4k.

## E    MORE EXPERIMENTS AND ANALYSIS

### E.1    REWRITING PROCESS

To investigate the differences in embedding distributions across texts of varying lengths, we use the Llama3 (*i.e.,* `Llama-3.1-8B-Instruct`) (Dubey et al., 2024) model to rewrite the texts. Specifically, we used two prompts, "*Please express the given text in one sentence. No more than 10 tokens. {{original text}}*" and "*Please use few sentences to summarize the given text. {{original text}}*", to summarize the texts. This rewriting allows the texts to retain the same semantics while having lower lengths. By studying the differences in texts of varying lengths, we conclud that the cause of the length collapse phenomenon is that longer texts cluster to each other in embedding space.

### E.2    DETAILS ON FIGURE 2

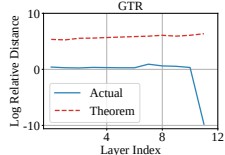 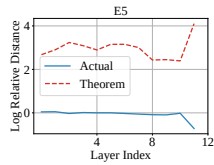

Figure 8: Visualization of the intensity of high-frequency components and their theoretical upper bounds. The setting is the same with Figure 2.

To verify Theorem 2, we illustrate the high-frequency intensity of each layer's output along with its theoretical upper limit. Our visualization is based on the official checkpoint of 12-layer ANCE,

GIST and BGE. We use a logarithmic scale for the purpose of a better view. Let $\boldsymbol{X}_l$ denote the output of the $l$-th layer. For red line, we directly calculate $\log(\|\mathcal{HC}\left[\boldsymbol{X}_{l+1}\right]\|_F / \|\mathcal{HC}\left[\boldsymbol{X}_l\right]\|_F)$ at each layer. In practice, models typically employ multi-head attention, so we replace $\|\boldsymbol{W}_V\|_2$ in Eqn. 2 with $\sigma_1 H$, where $\sigma_1 = \max_{h=1}^{H}\|\boldsymbol{W}_V^h\|_2$. For blue line, we obtain the coefficient $\sigma_1$ with respect to network parameters and apply the logarithmic scale. To summarize, Figure 2 imply an convergence rate, which is consistent with our Theorem 2. Figure 8 show the same trend as Figure 2, although the E5 model exhibits anomalies at deeper layers.

### E.3 DETAILS ON FIGURE 7

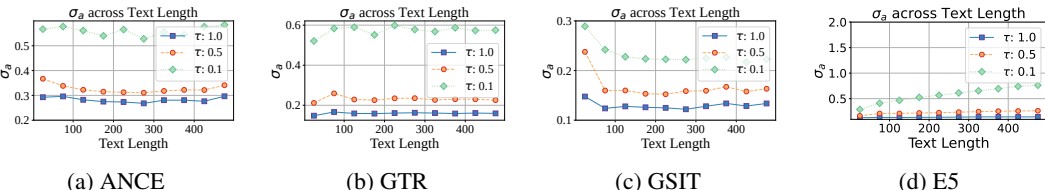

(a) ANCE         (b) GTR         (c) GSIT         (d) E5

Figure 9: $\sigma_a$ of $\mathcal{HC}\left[\boldsymbol{A}\right]$ before the last layer across different text length under different $\tau$ setting.

To verify Theorem 3, we visualize the value of $\sigma_a$ across different text length. Our visualization is based on the texts from NFCorpus. We sample 100 samples for each bin from 0 to 500 with a bin size of 50. The $\sigma_a$ value is computed as the average of the $\sigma_a$ based on the attention of all heads before the output of the final layer. To summarize, Figure 7 imply that $\sigma_a$ shows a decreasing trend as $n$ increases, which is consistent with our Theorem 3. Moreover, $\sigma_a$ also increases as $\tau$ decreases, further validating our proposed method, TempScale. Figure 9 also show the same trend as Figure 7 and the E5 model shows an increasing $\sigma_a$ with length, which may be due to anomalies in the deep layer attention patterns.

### E.4 MORE DISCUSSIONS ON CLASSIFICATION AND CLUSTERING TASKS

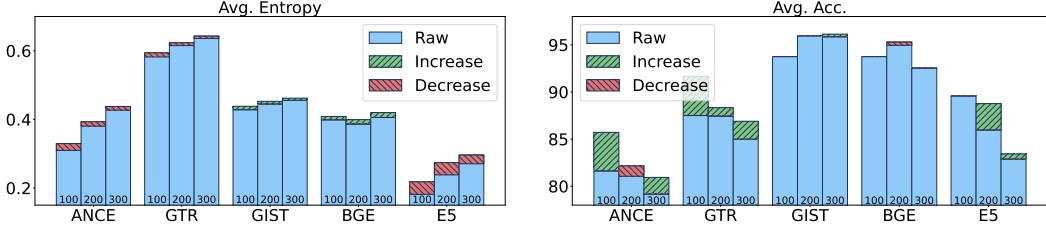

Figure 10: Probability entropy and classification accuracy of models across different length intervals on AmazonPolarity dataset. Each model has bars representing intervals of 100 in length, with 500 text samples per interval, covering a range from 0 to 300. Bars represent raw outputs, with green and red hatching indicating increases and decreases after TempScale, respectively.

For data grouping tasks like classification and clustering, in the case of $N$-class tasks, we can think of the model as learning $N$ classification boundaries. The farther the text embedding is from the boundary, the closer the output probability approaches 1 or 0. As shown in Figure 10 (left), we plot the entropy of the model output probabilities across different length intervals. The model outputs higher entropy for longer texts, which may be because the embeddings of longer texts are positioned closer to the center of the space as described in Figure 1b, resulting in a shorter distance to various classification boundaries. Meanwhile, in Figure 10 (right), we can also observe that accuracy and entropy follow the same trend: the model achieves higher accuracy when it has lower entropy. In other words, the model performs better if the text embeddings are farther from the boundary. After applying TempScale, a decreased entropy will result in increased accuracy. This further supports the relationship between entropy and accuracy in classification tasks. Moreover, if the model exhibits a more severe length collapse phenomenon, meaning a greater performance drop on longer texts,

the more performance improvement it experiences after applying TempScale. This suggests that one possible reason TempScale is effective on shorter text datasets is that it adjusts the distribution differences between texts of varying lengths. As shown in Figure 1c, after applying TempScale, the distribution of text embeddings across different length intervals becomes more uniform, which facilitates the model in learning length-agnostic parameters.

### E.5 MORE ANALYSIS ABOUT ASSUMPTION IN THEOREM 4

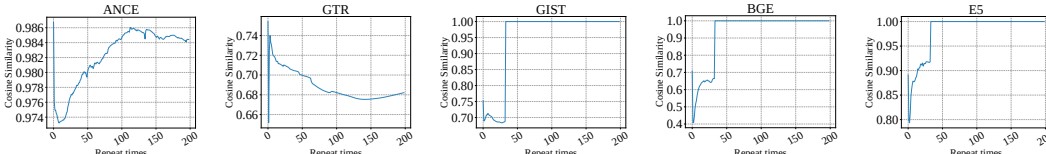

Figure 11: The cosine similarity of embeddings of texts generated by repeating word "dog" and "cat".

In Corollary 4, we hypothesize and verify that all natural language sequences tend to have a relatively consistent representation. As a result, different texts tend to exhibit consistent low-pass signals after losing high-frequency information, leading to ultimately consistent text embeddings. This leads to an increase in cosine similarity for longer texts. However, as shown in Figure 11, we repeat the word "dog" and "word" $n$ times and calculate the similarity between these two texts. The results show that even when the sequences do not have overlap, the text embeddings tend to converge to similar representations as the sequence length increases. This further demonstrates that length collapse causes completely different sequences to converge toward similarity.

### E.6 MORE RESULTS ON LONGER TEXTS

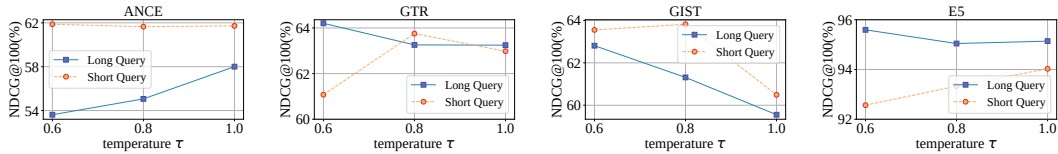

Figure 12: Results about performance difference between long query and short query across varying temperature $\tau$ on SummScreenFD dataset.

**Can the temperature be set based on the length of the text?** In the previous experiments, we use the same temperature $\tau$ for scaling all texts in the same task. However, our analysis indicates that texts of different lengths have varying filtering rates, so a natural idea is to use different temperatures for texts of different lengths. As shown in Figure 14, We plot the performance trend for texts under the same settings in Figure 10 as the temperature varies. The results indicate that a higher temperature is optimal for short texts, while a lower temperature is preferable for long texts, as confirmed by the results in Table 2. When performing retrieval tasks, we can also set different temperatures for queries and documents to achieve better performance. As shown in Figure 13, on the QMSum dataset, we can consistently achieve better performance by setting a lower temperature for queries. Moreover, as shown in Figure 15, compared to the QMSum dataset, SummScreenFD requires a lower temperature for scaling the document due to its longer length. This further supports the conclusion that longer texts require a lower temperature for scaling. In Figure 12, we observe a similar phenomenon across other models. Except for ANCE, the performance of other models on long queries decreases as the temperature decreases. This suggests that long queries require a lower temperature to mitigate the length collapse phenomenon. However, ANCE's performance degradation with decreasing temperature is likely attributable to its inherent limitations in processing long texts.

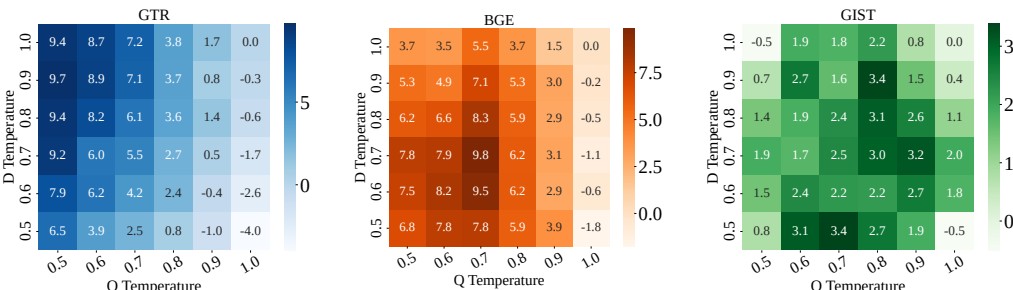

Figure 13: Relative performance compared to the raw results with varying query (Q Temperature) and document (D Temperature) temperatures using different models on QMSum.

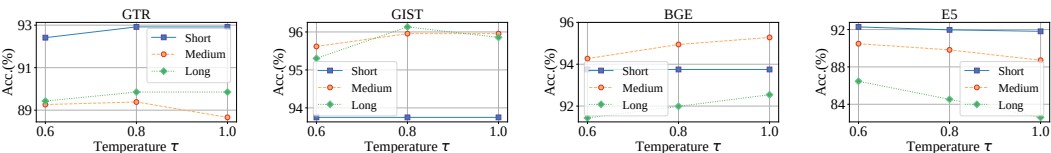

Figure 14: Results about performance difference between long query and short query across varying temperature $\tau$ on AmazonPolarity dataset.

## F    MORE DISCUSSIONS ABOUT TEMPSCALE

### F.1    RELATIONSHIP WITH CONTRASTIVE LEARNING METHODS

#### F.1.1    BACKGROUND AND MOTIVATION

Contrastive learning is a technique widely adopted to improve the anisotropy of the embedding space, which can reduce the high similarity between embeddings of random text samples across varied lengths. This anisotropy mitigation is achieved by maximizing distances among negative pairs while aligning positive pairs closely. However, while contrastive learning has demonstrated improvements in embedding quality across multiple applications, its effects on the length collapse phenomenon remain unexamined. This section explores how contrastive learning impacts length collapse, evaluating its contributions and limitations.

As shown in previous works Wang & Isola (2020), the InfoNCE loss, when scaled to a large number of negative samples, can be decomposed into two primary components: **Alignment** and **Uniformity**. Alignment ensures that positive pairs—texts with similar content—are close in the embedding space, while Uniformity spreads embeddings of negative pairs to prevent them from clustering excessively. Mathematically, as the number of negative samples $M \to \infty$, the normalized InfoNCE loss can be expressed as:

$$\lim_{M \to \infty} L(f, \tau) - \log M = -\frac{1}{\tau} E_{(x,x^+) \sim p_{\text{pos}}} \left[ f(x)^T f\left(x^+\right) \right] \qquad \text{Alignment}$$
$$+ E_{x \sim p_{\text{data}}} \left[ \log E_{x^- \sim p_{\text{data}}^-} \left[ e^{f(x)^T f\left(x^-\right)/\tau} \right] \right]. \qquad \text{Uniformity}$$

These properties make contrastive learning especially effective in tasks such as retrieval, where maximizing inter-sample variance is crucial.

#### F.1.2    WHY CONTRASTIVE LEARNING CANNOT FULLY ADDRESS LENGTH COLLAPSE

While contrastive learning alleviates high similarity issues across all text lengths, it does not entirely resolve the **length collapse** issue inherent in transformer-based models. Length collapse arises from self-attention's tendency to push embeddings for longer texts toward a concentrated representation space. This characteristic is unaffected by contrastive learning's alignment or uniformity mechanisms

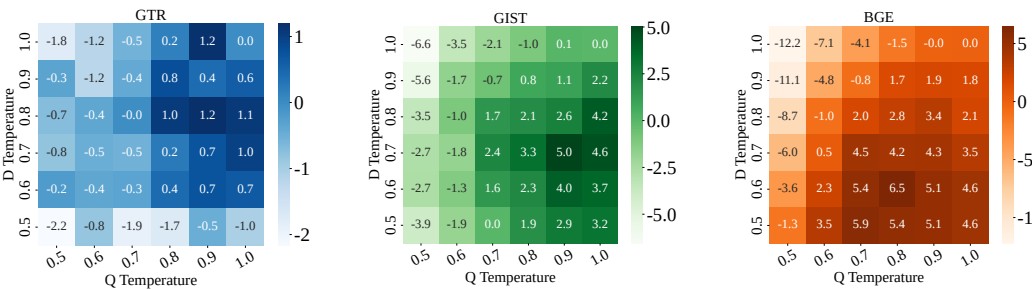

Figure 15: Relative performance compared to the raw results with varying query (Q Temperature) and document (D Temperature) temperatures using different model on SummScreenFD

because contrastive learning optimizes the relative positioning of positive and negative pairs rather than mitigating the length-induced clustering trend.

## F.2 COMPARISON WITH OTHER POST-PROCESSING TECHNIQUES

In addition to TempScale, several post-processing methods have been proposed to address high similarity in embeddings, such as the Flow Function Li et al. (2020) and Whitening Su et al. (2021). For comparative analysis, we implemented the last2avg version of the Flow Function, as the full flow version requires additional training, and the Whitening method is based on its original formulation. The results, summarized in Table 6, highlight that these methods do not perform as effectively in our specific application. These results indicate that Flow and Whitening, originally developed for standard BERT embeddings, are less effective with the fine-tuned transformer model used in this paper, which includes additional normalization layers. While TempScale reduces similarity in long-sequence embeddings, achieving distributional consistency across different sequence lengths is more critical for performance. Therefore, lowering similarity alone may not significantly improve downstream tasks.

Table 6: Performance comparison of different post-processing methods across various tasks

| Model | Rerank. | Summ. | Class. | Clust. | LongEmbdRetr. | STS | BeirRetr. | Avg. |
|---|---|---|---|---|---|---|---|---|
| **ANCE** | 49.09 | 29.58 | 55.27 | 33.04 | 34.02 | 66.32 | 36.87 | 43.45 |
| **+Flow** | 48.49 | 29.57 | 56.34 | 32.11 | 31.92 | 66.02 | 36.57 | 43.00 |
| **+Whitening** | 25.16 | 21.76 | 25.82 | 3.21 | 1.29 | 2.78 | 0.79 | 11.54 |
| **GTR** | 54.23 | 29.67 | 55.10 | 38.65 | 37.33 | 70.11 | 44.98 | 47.15 |
| **+Flow** | 37.49 | 28.09 | 41.98 | 21.87 | 7.65 | 35.55 | 1.35 | 24.85 |
| **+Whitening** | 24.63 | 20.69 | 25.71 | 3.06 | 1.35 | -3.02 | 0.63 | 10.44 |
| **GIST** | 58.55 | 31.14 | 64.75 | 44.77 | 38.21 | 75.61 | 52.77 | 52.26 |
| **+Flow** | 58.16 | 30.56 | 65.68 | 44.67 | 35.83 | 74.98 | 51.40 | 51.61 |
| **+Whitening** | 24.08 | 24.26 | 20.06 | 9.83 | 1.78 | -4.31 | 0.81 | 10.93 |
| **BGE** | 58.87 | 31.03 | 64.79 | 45.80 | 37.46 | 75.88 | 55.29 | 52.73 |
| **+Flow** | 58.48 | 30.73 | 64.77 | 45.28 | 36.95 | 74.43 | 54.52 | 52.16 |
| **+Whitening** | 24.75 | 19.61 | 16.50 | 4.23 | 2.39 | 1.60 | 0.47 | 9.94 |
| **E5-4K** | 53.12 | 30.58 | 61.72 | 41.01 | 56.01 | 71.77 | 47.22 | 51.63 |
| **+Flow** | 52.48 | 29.82 | 62.33 | 39.95 | 44.71 | 68.55 | 30.68 | 46.93 |
| **+Whitening** | 24.94 | 22.14 | 22.73 | 4.04 | 1.85 | 1.96 | 0.56 | 11.17 |

## F.3 COMPARISON WITH OTHER SIMILAR LONG-TEXT METHODS

To thoroughly evaluate our TempScale approach, we compare it with two related attention scaling methods: softmax $\left(\frac{\log n}{\tau\sqrt{d}}QK^T\right)V$ (Method1) (Chiang & Cholak, 2022) and YaRN (Peng et al.). In YaRN, the scaling formula for the attention matrix is given as softmax $\left(\frac{1}{\tau\sqrt{d}}QK^T\right)V$, where

$\frac{1}{\tau} = 0.1 \ln s + 1$, and $s = \frac{L'}{L}$, with $L'$ representing the extended context window length and $L$ the original context window length. These methods propose different scaling strategies, but they differ in their theoretical motivations, applications, and experimental outcomes.

While Method1, YaRN, and TempScale share a similar structural approach, they tackle different challenges. Method1 is for binary classification tasks, specifically determining if the first string in a sequence of binary strings is a '1'. It focuses on resolving straightforward decision-making problems in binary data. YaRN, by contrast, extends the context window of LLMs without requiring retraining, particularly adjusting the attention mechanism to handle larger contexts. TempScale addresses length collapse in embedding models for long texts, improving performance without retraining by preserving distinct representations as text length increases. Moreover, Method1 offers a solution using a specially designed Transformer example, but its motivation is limited by the specificity of this example. In contrast, YaRN introduces an empirically derived scaling formula. TempScale is based on a rigorous low-pass filtering analysis, giving it a strong theoretical foundation and an intuitive explanation.

Table 7: Average main metric on MTEB and LongEmbd across Method1 and YaRN.

| Model | Rerank. | Summ. | Class. | Clust. | LongEmbdRetr. | STS | BeirRetr. | Avg. |
|---|---|---|---|---|---|---|---|---|
| ANCE | 49.09 | 29.58 | 55.27 | 33.04 | 34.02 | 66.32 | 36.87 | 43.45 |
| +Method1 | 41.80 | 29.21 | 48.21 | 20.72 | 6.23 | 53.40 | 7.74 | 29.61 |
| +YaRN($\lambda = 0.0001$) | 49.09 | 29.58 | 55.62 | 33.01 | 34.02 | 66.32 | 36.86 | 43.50 |
| +YaRN($\lambda = 0.001$) | 49.08 | 29.57 | 55.62 | 33.05 | 33.96 | 66.32 | 36.87 | 43.49 |
| +YaRN($\lambda = 0.01$) | 49.10 | 29.31 | 55.65 | 32.95 | 33.92 | 66.27 | 36.87 | 43.44 |
| +YaRN($\lambda = 0.1$) | 48.95 | 29.30 | 55.45 | 32.65 | 32.31 | 65.72 | 35.77 | 42.88 |
| +YaRN($\lambda = 1$) | 45.71 | 29.28 | 52.80 | 26.16 | 10.59 | 60.15 | 13.34 | 34.00 |
| GTR | 54.23 | 29.67 | 55.10 | 38.65 | 37.33 | 70.11 | 44.98 | 47.15 |
| +Method1 | 38.12 | 28.44 | 40.34 | 14.46 | 3.61 | 47.09 | 0.82 | 24.70 |
| +YaRN($\lambda = 0.0001$) | 54.23 | 29.68 | 55.06 | 38.30 | 37.42 | 70.11 | 44.98 | 47.11 |
| +YaRN($\lambda = 0.001$) | 54.23 | 29.71 | 55.08 | 38.29 | 37.56 | 70.11 | 44.97 | 47.13 |
| +YaRN($\lambda = 0.01$) | 54.23 | 29.71 | 55.04 | 38.63 | 37.39 | 70.06 | 45.00 | 47.15 |
| +YaRN($\lambda = 0.1$) | 54.12 | 29.34 | 54.86 | 34.07 | 36.53 | 69.86 | 43.06 | 45.98 |
| +YaRN($\lambda = 1$) | 55.19 | 29.00 | 47.27 | 19.06 | 3.99 | 61.87 | 1.46 | 31.12 |
| GIST | 58.55 | 31.14 | 64.75 | 44.77 | 38.21 | 75.61 | 52.77 | 52.26 |
| +Method1 | 58.64 | 28.26 | 51.05 | 28.11 | 5.42 | 62.43 | 7.51 | 34.49 |
| +YaRN($\lambda = 0.0001$) | 58.55 | 31.14 | 64.26 | 44.75 | 38.21 | 75.61 | 52.78 | 52.19 |
| +YaRN($\lambda = 0.001$) | 58.55 | 31.12 | 64.28 | 44.75 | 38.20 | 75.61 | 52.78 | 52.18 |
| +YaRN($\lambda = 0.01$) | 58.53 | 31.26 | 64.24 | 44.78 | 38.05 | 75.60 | 52.79 | 52.18 |
| +YaRN($\lambda = 0.1$) | 58.45 | 30.36 | 63.86 | 44.62 | 37.34 | 75.31 | 52.10 | 51.72 |
| +YaRN($\lambda = 1$) | 55.75 | 26.76 | 58.79 | 36.95 | 11.85 | 69.00 | 23.59 | 40.38 |
| BGE | 58.87 | 31.03 | 64.79 | 45.80 | 37.46 | 75.88 | 55.29 | 52.73 |
| +Method1 | 37.78 | 29.17 | 37.80 | 14.36 | 2.10 | 43.72 | 0.84 | 23.68 |
| +YaRN($\lambda = 0.0001$) | 58.86 | 31.04 | 64.78 | 45.77 | 37.45 | 75.88 | 55.29 | 52.72 |
| +YaRN($\lambda = 0.001$) | 58.86 | 30.96 | 64.78 | 45.75 | 37.47 | 75.87 | 55.28 | 52.71 |
| +YaRN($\lambda = 0.01$) | 58.85 | 31.02 | 64.73 | 45.61 | 37.26 | 75.86 | 55.22 | 52.65 |
| +YaRN($\lambda = 0.1$) | 58.86 | 30.90 | 64.51 | 45.19 | 36.55 | 75.55 | 54.96 | 52.36 |
| +YaRN($\lambda = 1$) | 52.65 | 29.09 | 50.51 | 25.50 | 2.36 | 64.20 | 2.14 | 32.35 |
| E5-4K | 53.12 | 30.58 | 61.72 | 41.01 | 56.01 | 71.77 | 47.22 | 51.63 |
| +Method1 | 40.90 | 24.11 | 42.85 | 13.64 | 3.24 | 41.62 | 1.05 | 23.92 |
| +YaRN($\lambda = 0.0001$) | 53.12 | 30.57 | 61.78 | 40.77 | 56.01 | 71.77 | 47.22 | 51.61 |
| +YaRN($\lambda = 0.001$) | 53.11 | 30.55 | 61.78 | 40.75 | 55.98 | 71.77 | 47.22 | 51.59 |
| +YaRN($\lambda = 0.01$) | 53.07 | 30.42 | 61.73 | 40.61 | 55.53 | 71.71 | 47.21 | 51.47 |
| +YaRN($\lambda = 0.1$) | 52.64 | 30.20 | 61.51 | 40.19 | 48.39 | 70.68 | 46.36 | 50.00 |
| +YaRN($\lambda = 1$) | 45.00 | 29.01 | 50.81 | 17.99 | 2.91 | 57.27 | 1.46 | 29.21 |

To effectively compare the performance of the above methods, we tested them on the MTEB benchmark. Since we are not modifying the context window length, we adapt YaRN as softmax $\left( \frac{\tau}{\sqrt{d}} Q K^T \right) V$, with $\tau = \lambda \log n + 1$. We explore values for $\lambda$ in the range $\{0.0001, 0.001, 0.01, 0.1, 1\}$. This setup is reasonable, as our findings indicate that longer texts generally benefit from a smaller temperature scale. The experimental results are as shown in Table 7.

The experimental results indicate that neither Method1 nor YaRN effectively adapts to the embedding model scenario. (Although YaRN shows slight improvement when the $\lambda$ value is small, in this

case, Method 2 degenerates into TempScale.) Some potential reasons are as follows: A plausible explanation is that while both methods perform finer scaling on the attention matrix, applying different temperature adjustments across varying text lengths may lead to embeddings from different lengths falling into distinct distributions, which is unfavorable for downstream tasks. Moreover, in generation tasks, Method1 and YaRN succeed likely because of differences in output requirements. For these tasks, the model outputs a probability distribution and samples from it. Minor perturbations generally don't affect the token output significantly; even if one sequence's token distribution changes, it doesn't impact the output of other sequences. In contrast, any substantial change in a single embedding for an embedding model can directly affect the entire downstream performance. For instance, in classification tasks, a classifier model relies on embeddings as input, and in retrieval tasks, a change in embedding impacts document ranking. Overall, for embedding tasks, simply applying a single temperature adjustment better maintains the overall embedding distribution, helping mitigate length collapse and achieve better results across various downstream applications.

## G  LENGTH COLLAPSE IN LLM-BASED EMBEDDING MODELS

To explore whether length collapse also occurs in long-context LLMs, we select three widely used LLM-based embedding models from the MTEB benchmark—bge-multilingual-gemma2 (Chen et al., 2024), NV-Embed-v2 (Lee et al., 2024), and e5-mistral-7b-instruct (Wang et al., 2022a)—all of which rank within the top 20 on the MTEB leaderboard. We conduct experiments using three commonly used long-text datasets, including nq and hotpotqa in LongRAG (Jiang et al., 2024) and LongAlpaca-12k (Chen et al.), which are selected based on relevance to the keyword "long" and chosen for their evenly distributed text lengths. Specifically, we analyze shifts in embedding space across different text length intervals by calculating the average Euclidean distance of each embedding from the central embedding (the mean of all embeddings) and computing cosine similarity between each pair of embeddings as in Table 8 and Table 9.

Table 8: Euclidean distance results for different datasets and LLM-based embedding models.

| Dataset | Model | 0-1000 | 1000-2000 | 2000-3000 |
|---|---|---|---|---|
| nq | bge-multilingual-gemma2 | 0.94 | 0.92 | 0.91 |
| | NV-Embed-v2 | 0.98 | 0.91 | 0.89 |
| | e5-mistral-7b-instruct | 0.68 | 0.64 | 0.64 |
| hotpotqa | bge-multilingual-gemma2 | 0.93 | 0.88 | 0.87 |
| | NV-Embed-v2 | 0.95 | 0.84 | 0.79 |
| | e5-mistral-7b-instruct | 0.70 | 0.61 | 0.57 |
| LongAlpaca-12k | bge-multilingual-gemma2 | 0.71 | 0.48 | 0.49 |
| | NV-Embed-v2 | 0.89 | 0.86 | 0.87 |
| | e5-mistral-7b-instruct | 0.69 | 0.53 | 0.53 |

Table 9: Pair cosine similarities results for different datasets and LLM-based embedding models.

| Dataset | Model | 0-1000 | 1000-2000 | 2000-3000 |
|---|---|---|---|---|
| nq | bge-multilingual-gemma2 | 0.39 | 0.42 | 0.45 |
| | NV-Embed-v2 | 0.32 | 0.47 | 0.59 |
| | e5-mistral-7b-instruct | 0.75 | 0.77 | 0.78 |
| hotpotqa | bge-multilingual-gemma2 | 0.51 | 0.52 | 0.59 |
| | NV-Embed-v2 | 0.48 | 0.59 | 0.69 |
| | e5-mistral-7b-instruct | 0.75 | 0.81 | 0.85 |
| LongAlpaca-12k | bge-multilingual-gemma2 | 0.76 | 0.90 | 0.89 |
| | NV-Embed-v2 | 0.48 | 0.65 | 0.68 |
| | e5-mistral-7b-instruct | 0.78 | 0.87 | 0.87 |

The experimental results show that as text length increases, the embeddings from different LLM-based embedding models exhibit a gradual convergence trend, with the average Euclidean distance between embeddings decreasing and pairwise cosine similarity increasing. This indicates that even mainstream long-context LLM embedding models tend to experience embedding convergence (Length Collapse) and reduced distinctiveness in long-text processing due to the low-pass filtering effect.

