# OpenReview forum: "Length-Induced Embedding Collapse in Transformer-based Models"
_ICLR.cc/2025/Conference — Submitted to ICLR 2025_

### Official Review · Reviewer_hoZs · 2024-10-28

**Soundness:** 3
**Presentation:** 3
**Contribution:** 2
**Rating:** 6
**Confidence:** 4

**Summary:**

This paper describes the “length collapse” problem in text embedders. It extends previous work treating the self-attention mechanism in the Transformer as a low-pass filter by demonstrating that long sequences increase the attenuation rate of the low-pass filter effect of the self-attention mechanism. Then, the work introduces an additional temperature coefficient that results in a higher low-filter decay rate to mitigate the length collapse problem.

**Strengths:**

1. The paper is well organized and understandable, and the experiments are adequate.

2. This work attempts to relate the Transformer's self-attention module to the “length collapse” problem in embedding space, which is novel, as most of the related work has only investigated the related problems through empirical studies or manipulations on embedding space.

3. The proposed TempScale is orthogonal to existing methods, can be applied to various state-of-the-art models by plug-and-play, and is training-free.

**Weaknesses:**

1. The "transformer-based Models" in the title are too big because the analysis is only related to the self-attention mechanism, not to the other architectures in the Transformer.

2. In general, TempScale is a post-processing method for text embedding that solves the problem of too much similarity in long sequence embedding. Still, I don't see any comparison between TempScale and some post-processing baseline methods, e.g., Flow Function [1], and Whitening [2], to solve the similarity problem.

3. The improvement of TempScale on each model is inconsistent, and the main improvement does not come from the long text part (as shown in Figure 6), so more explanation and exploration are necessary.

4. The relevant work [3] and the important long-context embedder [4] have been overlooked.

[1] On the Sentence Embeddings from Pre-trained Language Models. EMNLP2020

[2] Whitening Sentence Representations for Better Semantics and Faster Retrieval. Arxiv

[3] Length is a Curse and a Blessing for Document-level Semantics. EMNLP2023

[4] mGTE: Generalized Long-Context Text Representation and Reranking Models for Multilingual Text Retrieval. EMNLP2024

**Questions:**

All embedding models studied by the authors have been fine-tuned based on contrastive learning; however, this technique seems to have been deliberately avoided in this work. Contrastive learning was first introduced to improve anisotropy in the embedding space of PLMs, and one manifestation of anisotropy is high similarity between random texts. In other words, without contrastive learning, the similarity between two short text embeddings, not only long texts, is also high.

Therefore, several questions need to be explained:

(1) How to explain the change of embedding space before and after contrastive learning from the filter perspective;

(2) Why there is still the problem of “length collapse” after contrastive learning, and where the contrastive learning learning is not good enough;

(3) The introduction of long text in contrastive learning should make the model perform better on long text, so how can we understand this phenomenon from the filter perspective?

The contribution of this work is very limited if the authors' study is isolated and not linked to contrastive learning. In practice, one would be more inclined to introduce more long text for contrastive fine-tuning than to use a post-processing method such as TempScale, since clearly the former would be more effective.

---

> ### Author Response · Authors · 2024-11-19
> **Author Comment: "Transformer-based" model in title and more post-processing method comparison**
>
> We are grateful for the reviewer's recognition and insightful comments. We will address each point below.
>
> ### W1
> > The "transformer-based Models" in the title are too big because the analysis is only related to the self-attention mechanism, not to the other architectures in the Transformer.
>
> Thank you for your valuable feedback on our research! Below is a further explanation of your questions:
>
> - **Regarding the classification of embedding models:** Indeed, some papers [1][2] classify embedding models from the perspective of model architecture, typically dividing them into models based on different architectures such as Transformer, LSTM, etc. In light of this classification, we chose to include “Transformer” in the title to emphasize that our study specifically focuses on this model architecture and its key self-attention mechanism.
>
> - **Background of the title selection**: We understand your concern about the term “transformer-based models” in the title, as it may suggest that our research involves the entire Transformer architecture rather than specific components. **However, we use the term 'Transformer-based' to describe the phenomenon rather than the cause of the phenomenon.** Based on prior classifications, our study focuses on the length collapse phenomenon observed within Transformer-based models, which may lead to performance degradation on long texts. Our goal is to understand this phenomenon more deeply, and we ultimately identified the attention mechanism as a contributing factor. Therefore, we use the term “Transformer-based models” to indicate that this phenomenon specifically affects this model class.
>
> - **Further discussion on the Transformer module:** Additionally, in Section 3.2, "Discussion," we examine other components of the Transformer architecture and their impact on this phenomenon. **Our discussions indicate that components such as LayerNorm and FFN help mitigate oversmoothing as layers increase, but this effect is not dependent on sequence length.** Thus, while our focus is on self-attention, the overall analysis applies broadly to the Transformer model.
>
> If you have any further suggestions, we will consider modifying the title accordingly based on your feedback.
>
> ### W2
> > In general, TempScale is a post-processing method for text embedding that solves the problem of too much similarity in long sequence embedding. Still, I don't see any comparison between TempScale and some post-processing baseline methods, e.g., Flow Function [1], and Whitening [2], to solve the similarity problem.
>
> Thank you for suggesting additional comparison methods. We implemented these methods for a performance comparison, specifically using the last2avg version from the Flow Function paper, as the flow version requires training. We also implemented the Whitening method based on its original paper.
>
>
> |Model|Reranking|Summarization|Classification|Clustering|LongEmbdRetrival|STS|BeirRetrival|avg.|
> |:-:|:-:|:-:|:-:|:-:|:-:|:-:|:-:|:-:|
> |ANCE|49.09|29.58|55.27|33.04|34.02|66.32|36.87|43.45|
> |+Flow|48.49|29.57|56.34|32.11|31.92|66.02|36.57|43.00|
> |+Whitening|25.16|21.76|25.82|3.21|1.29|2.78|0.79|11.54|
> |GTR|54.23|29.67|55.10|38.65|37.33|70.11|44.98|47.15|
> |+Flow|37.49|28.09|41.98|21.87|7.65|35.55|1.35|24.85|
> |+Whitening|24.63|20.69|25.71|3.06|1.35|-3.02|0.63|10.44|
> |GIST|58.55|31.14|64.75|44.77|38.21|75.61|52.77|52.26|
> |+Flow|58.16|30.56|65.68|44.67|35.83|74.98|51.40|51.61|
> |+Whitening|24.08|24.26|20.06|9.83|1.78|-4.31|0.81|10.93|
> |BGE|58.87|31.03|64.79|45.80|37.46|75.88|55.29|52.73|
> |+Flow|58.48|30.73|64.77|45.28|36.95|74.43|54.52|52.16|
> |+Whitening|24.75|19.61|16.50|4.23|2.39|1.60|0.47|9.94|
> |E5-4K|53.12|30.58|61.72|41.01|56.01|71.77|47.22|51.63|
> |+Flow|52.48|29.82|62.33|39.95|44.71|68.55|30.68|46.93|
> |+Whitening|24.94|22.14|22.73|4.04|1.85|1.96|0.56|11.17|
>
> The results indicate that these two post-processing methods do not perform well, for two main reasons:
>
> - **1.Compatibility with Model Architecture**: These methods were originally proposed for standard BERT embeddings, while our model has been fine-tuned for downstream tasks (e.g., matching and classification) and includes additional normalization layers. Applying these post-processing methods can disrupt the semantic information in the embeddings.
> - **2.Relevance to Distributional Consistency**: While high similarity in long text embeddings is a phenomenon we observed, the primary factor impacting downstream task performance is **the distributional inconsistency between short and long texts, not just high similarity**. For example, in retrieval tasks, the ranking of document scores relative to a query matters most. If three documents initially score 0.999, 0.998, and 0.997, post-processing may adjust these scores to 0.9, 0.8, and 0.7, reducing similarity but keeping the ranking intact. Thus, reducing similarity alone doesn’t improve downstream performance.

---

> > ### Author Response · Authors · 2024-11-19
> > **Author Comment: Inconsistent improvement and relavant work and model**
> >
> > ### W3
> > > The improvement of TempScale on each model is inconsistent, and the main improvement does not come from the long text part (as shown in Figure 6), so more explanation and exploration are necessary.
> >
> > Thank you for your feedback.
> >
> > - **For Consistency improvement**： We recognize that TempScale's improvements vary across models due to differences in architecture. For example, GTR is based on T5, ANCE on RoBERTa, and both GIST and BGE on BERT, which leads to similar improvement patterns in GIST and BGE. Additionally, for simplicity, we used the same temperature setting across models in our main table, which may have impacted performance on some tasks. Setting optimal temperatures per task could yield more consistent improvements across models.
> >
> > - **For Short-text Performance**: **Furthermore, as for Figure 6 showing greater improvement on short texts than long texts, this is due to the distributional properties of embeddings.** For example, in MTEB’s KNN classifier, short texts are more dispersed in the embedding space, while long texts, due to length collapse, cluster toward the center. During KNN optimization, long texts, being closer to the embedding center, influence voting results more strongly. By applying TempScale (as shown in Figure 1c), short and long texts become more aligned in distribution, enabling short texts to have a greater influence and leading to a larger improvement in performance on short texts.
> >
> > - **For Long-Text Performance**: Finally, Figures 7, 8, and 9 demonstrate that TempScale helps mitigate length collapse, providing more contextual representations for long texts and enhancing model performance on these examples.
> >
> > ### W4
> > > The relevant work [3] and the important long-context embedder [4] have been overlooked.
> >
> > Thank you for providing your work and model.
> > For the relevant work, our study differs as follows:
> >
> > - **Context Differences:** While the relevant work focuses on high similarity in repeated text segments, our study identifies this phenomenon across all natural language sequences.
> > - **Mechanism Explanation:** [3] attributes the similarity issue in repeated texts to reduced attention to the [CLS] token. In contrast, **we rigorously analyze this from a low-pass filtering perspective,** showing that long text embeddings cluster in a narrower space, increasing similarity between long texts compared to shorter ones. Additionally, we reveal that the distributional inconsistency between long and short text embeddings contributes to the reduced performance of long texts in downstream tasks.
> >
> > **Limitations with the important long-context embedder Model (GTE):** Regarding the GTE model, we initially considered it as a base model for our main experiment. However, as GTE relies on a custom-defined model, we were unable to access the code for its attention module, making it incompatible with our TempScale method. To address the absence of a long-text base model, we used the e5-4k-base model instead, which can handle texts up to 4k tokens and has demonstrated strong performance on the LongEmbed Benchmark. Furthermore, we still observed the Length Collapse phenomenon in the GTE model (Alibaba-NLP/gte-base-en-v1.5). Specifically, We conducted experiments using three commonly used long-text datasets, including LongRAG and LongAlpaca-12k from HuggingFace, which were selected based on relevance to the keyword "long" and chosen for their evenly distributed text lengths. We analyzed distribution shifts in embedding space across different text length intervals, calculating the average Euclidean distance of these embeddings from the central embedding and the cosine similarity between each pair of embeddings.
> >
> > #### Euclidean distance
> > |Dataset | 0-1000 | 1000-2000 | 2000-3000 |
> > |:-:|:-:|:-:|:-:|
> > | LongRAG/nq_corpus|19.6| 19.23 | 18.94 |
> > | LongRAG/hotpot_qa_corpus |20.02 | 17.03 | 16.39 |
> > |LongAlpaca-12k|19.72 | 17.52 | 17.45 |
> >
> > #### Pair Cosine Similarity
> > |Dataset | 0-1000 | 1000-2000 | 2000-3000 |
> > |:-:|:-:|:-:|:-:|
> > | LongRAG/nq_corpus| 0.48 |0.47 |0.50 |
> > | LongRAG/hotpot_qa_corpus | 0.55 |0.65 |0.72 |
> > |LongAlpaca-12k| 0.48 |0.47 |0.50 |
> >
> > The experimental results show that as text length increases, embeddings from the GTE model display a convergence trend, with decreasing average Euclidean distances between embeddings and increasing pairwise cosine similarities. This suggests that even mainstream long-context embedding models experience embedding convergence (Length Collapse).
> >
> > Furthermore, we also explored whether such Length Collapse affects the model's performance. To investigate this, we tested the GTE model on the IMDB dataset across different text length intervals, and the results are as follows:
> >
> > #### Acc on ImbdClassification
> > |0-300|300-600|600-900|900-1200|
> > |:-:|:-:|:-:|:-:|
> > |0.94|0.93|0.91|0.88|
> >
> > The results indicate that, despite the GTE model having a longer context window, it still experiences a performance decline on longer texts.

---

> > > ### Author Response · Authors · 2024-11-19
> > > **Author Comment: Discussions on contrastive learning and its relation to Length Collapse (1/2)**
> > >
> > > ### Questions:
> > > > All embedding models studied by the authors have been fine-tuned based on contrastive learning; however, this technique seems to have been deliberately avoided in this work. Contrastive learning was first introduced to improve anisotropy in the embedding space of PLMs, and one manifestation of anisotropy is high similarity between random texts. In other words, without contrastive learning, the similarity between two short text embeddings, not only long texts, is also high.
> > > >
> > > > Therefore, several questions need to be explained:
> > > >
> > > > (1) How to explain the change of embedding space before and after contrastive learning from the filter perspective;
> > > >
> > > > (2) Why there is still the problem of “length collapse” after contrastive learning, and where the contrastive learning learning is not good enough;
> > > >
> > > > (3) The introduction of long text in contrastive learning should make the model perform better on long text, so how can we understand this phenomenon from the filter perspective?
> > > >
> > > > The contribution of this work is very limited if the authors' study is isolated and not linked to contrastive learning. In practice, one would be more inclined to introduce more long text for contrastive fine-tuning than to use a post-processing method such as TempScale, since clearly the former would be more effective.
> > >
> > >
> > > Thank you for your insightful questions, which address some of the core assumptions and mechanisms underlying our study. Below, I will address each of your points in detail, clarifying how our work relates to contrastive learning and the inherent characteristics of long-text embeddings.
> > >
> > > Our study focuses on a fundamental trend in the self-attention mechanism, where embeddings of long texts tend to cluster more tightly in the embedding space than those of short texts. While contrastive learning can reduce similarity across text embeddings (irrespective of length), **this clustering phenomenon with long texts persists regardless of whether contrastive learning is applied.** Hence, the issue is intrinsic to self-attention and exists universally across models.
> > >
> > > Additionally, we selected models that have already been fine-tuned with contrastive learning to ensure alignment with real-world scenarios, where such fine-tuning has become standard. However, examining the impact of contrastive learning on “length collapse” through a frequency domain perspective could indeed enrich our analysis. I will now address each of your questions individually.

---

> > > > ### Author Response · Authors · 2024-11-19
> > > > **Author Comment:Discussions on contrastive learning and its relation to Length Collapse (2/2)**
> > > >
> > > > 1. **Regarding the change in embedding space before and after contrastive learning**:
> > > >     Contrastive learning introduces two key objectives: **alignment** (bringing positive pairs closer) and **uniformity** (pushing negative pairs further apart). When the number of negatives $M$ grows large, the normalized InfoNCE loss simplifies to:
> > > >
> > > >     $$
> > > >     \lim _ {M \to \infty} L(f, \tau) - \log M = -\frac{1}{\tau} \mathbb{E} _ {(x, x^+) \sim p _ {pos}} [f(x)^T f(x^+)] \quad \text{Alignment}
> > > >     $$
> > > >
> > > >     $$ + \mathbb{E} _ {x \sim p _ {data}} \left[ \log \mathbb{E} _ {x^- \sim p _ {data}^-} \left[ e^{f(x)^T f(x^-) / \tau} \right] \right] \quad \text{Uniformity}
> > > >     $$
> > > >
> > > >     In this formula, **alignment** measures the similarity between anchor-positive pairs, while **uniformity** assesses anchor-negative distances. This second term (uniformity) pushes embeddings apart and helps counteract high similarity, creating more spread-out embeddings in the space.
> > > >
> > > >     From a **filter perspective**, better uniformity suggests that contrastive learning retains more high-frequency information in the embedding space, reducing the “low-pass” effect where the model filters out details. This explains why contrastive learning often improves embedding model performance in downstream tasks.
> > > >
> > > > 2. **Addressing the persistence of "length collapse" after contrastive learning**:
> > > >     Length collapse results from the self-attention mechanism’s behavior, where long texts tend to have embeddings that naturally concentrate in a narrower region. While contrastive learning can enhance alignment and uniformity, it is not specifically designed to prevent this tendency of long texts clustering more tightly. Contrastive learning focuses on increasing relative distances between positive and negative samples, but this does not change the self-attention mechanism’s intrinsic compression effect on longer texts. Thus, contrastive learning cannot fully resolve the length collapse issue.
> > > >
> > > > 3. **On the impact of contrastive learning on long-text embeddings**:
> > > >     In theory, contrastive learning should enhance the model’s alignment capabilities and thus improve its handling of both short and long texts. However, a challenge arises with long texts due to a **distributional mismatch** between embeddings of short and long texts. From a filtering perspective, longer texts tend to retain more low-frequency information, while shorter texts capture relatively higher frequencies. This discrepancy in frequency content creates a mismatch, leading to differences in performance for long versus short texts.
> > > >
> > > > Your questions have helped highlight that the performance drop on long texts is not merely due to high similarity within embeddings but also due to this distributional mismatch. This inconsistency introduces length-based biases in downstream tasks, meaning that focusing solely on similarity measures does not fully explain the performance decline on long texts.
> > > >
> > > > In previous studies, contrastive learning has often been applied to improve embedding quality, mainly by enhancing alignment and reducing embedding similarity. While contrastive learning can alleviate high similarity in Transformer-based models (and increase anisotropy), **recent findings [4] suggest that downstream tasks have varying anisotropy needs: clustering tasks benefit from low anisotropy, while retrieval tasks require higher anisotropy.** This implies that methods aimed at reducing high similarity (such as contrastive learning or post-processing methods like Flow Functions or Whitening) may not improve performance across all tasks.
> > > >
> > > > In conclusion, while contrastive learning improves embedding quality for certain tasks, it does not fully address performance degradation on long texts due to the fundamental distributional mismatch. We appreciate your detailed questions and will enhance the discussion around contrastive learning in our paper.
> > > >
> > > > Thank you for your valuable feedback, especially regarding the theoretical and experimental sections. Your suggestions have helped clarify essential issues in our research and significantly improved the quality of our work. If you have further questions or suggestions, we are eager to address them to ensure a comprehensive examination of all aspects of our study.
> > > >
> > > >
> > > > [1] A survey on contextual embeddings. arXiv2020.
> > > >
> > > > [2] Dense text retrieval based on pretrained language models: A survey. TOIS2024.
> > > >
> > > > [3] Understanding contrastive representation learning through alignment and uniformity on the hypersphere. ICML2020.
> > > >
> > > > [4] Effective post-training embedding compression via temperature control in contrastive training.

---

> > > > > ### Comment · Reviewer_hoZs · 2024-11-21
> > > > > **Thanks to the authors for their feedback**
> > > > >
> > > > > The author’s explanation has clarified some of my questions, particularly regarding the impact of the “length collapse” phenomenon on clustering and KNN-based classification. However, I still have concerns about its effect on the retrieval task discussed in the paper. Specifically, why does “length collapse” impair performance, and how does TempScale increase the average ranking of long documents? The problem is that the current analysis primarily focuses on documents of similar lengths rather than examining the interaction between short and long documents, which is crucial for asymmetric text-matching tasks.
> > > > >
> > > > > Additionally, I suggest analyzing some specific downstream tasks more comprehensively. Constructing reproducible toy models for targeted analysis could make the findings more intuitive. Reporting results directly on MTEB is less insightful, as the performance gains are minimal for certain tasks (e.g., Rerank), and the current analysis does not adequately address them; for others (e.g., Summary), the improvements lack rigorous interpretability.
> > > > >
> > > > > In summary, the experimental section still has significant room for improvement to strengthen this work's robustness. As such, I will maintain my current evaluation score.

---

> > > > > > ### Author Response · Authors · 2024-11-24
> > > > > > **Thanks for Reviewer Comments**
> > > > > >
> > > > > > Thank you very much for your suggestions.
> > > > > >
> > > > > > Our research primarily focuses on identifying the phenomenon of Length Collapse and attempting to explain it using Fourier analysis. Additionally, to address Length Collapse, we explored using TempScale for correction. Results on the MTEB benchmark demonstrate that mitigating Length Collapse can help alleviate performance degradation of mainstream embedding models on long texts.
> > > > > >
> > > > > > Howerver, understanding how Length Collapse impacts downstream task performance is crucial for proposing effective methods and further explains why TempScale is effective. **Thus, in Appendix F, we provide additional discussion and experimental validation regarding how Retrieval and STS are affected by Length Collapse, and we further discuss how TempScale alleviates the impact of Length Collapse in these two types of matching tasks.**
> > > > > >
> > > > > > Firstly, tasks that compute relevance scores based on matching can be divided into two categories: **Retrieval and STS**. In the MTEB benchmark, Retrieval tasks include retrieval, long-embed retrieval, and rerank, where there are significant length differences between the two texts being matched. The other category includes STS and summarization, where the two texts have similar lengths.
> > > > > >
> > > > > > Specifically, in Retrieval tasks, longer documents tend to occupy a limited central space with high similarity to all embeddings, while shorter, peripheral queries have more contextual variance, which can make shorter, noisier documents appear falsely relevant due to their larger representational space. The analysis on the NFCorpus dataset reveals that short relevant documents often rank at the extremes due to contextual variance, while long documents rank more uniformly but higher overall due to their higher similarity; TempScale improves long document rankings by enhancing their contextual distribution. In STS tasks, length collapse causes long-text embeddings to cluster closely, leading unrelated pairs to appear overly similar; TempScale addresses this by expanding long-document embedding space, enhancing separation for unrelated pairs while preserving high similarity for related ones.
> > > > > >
> > > > > > Finally, the poor performance on Rerank may be due to the smaller document candidate list (Approximately 10) compared to Retrieval(Approximately 1000), resulting in a smaller range of variation. In the Summary task, the performance fluctuations are more significant because there is only one dataset.
> > > > > >
> > > > > > Thank you again for your valuable suggestions. They have really helped us improve our paper!

---

> > > > > > > ### Comment · Reviewer_hoZs · 2024-11-25
> > > > > > > **Thanks to the authors for the timely feedback**
> > > > > > >
> > > > > > > Thanks to the authors' timely feedback, the added appendix section has greatly improved the quality of the paper, while this responsible attitude is impressive. Therefore, I raise my rating to 6.0.
> > > > > > >
> > > > > > > By the way, I think the authors' additions to the appendix are more insightful than the experiment section in the main text, so I strongly recommend that a portion of the appendix be reorganized into the main text section. Good luck!

---

> > > > > > > > ### Author Response · Authors · 2024-11-25
> > > > > > > > **Thanks for Reviewer Comments**
> > > > > > > >
> > > > > > > > Thank you for recognizing our work and increasing the score! We greatly appreciate your ongoing suggestions and positive interactions.
> > > > > > > >
> > > > > > > > In the revised version of the paper, we have moved the toy example and discussion on performance decline caused by Length Collapse from the appendix to Section 3, placing it between the analysis of Length Collapse and the proposed solution, TempScale. This change aims to better help readers grasp the core issue and improve the paper's quality. Additionally, we have streamlined the analysis in the experimental section, as Section 3 now provides extensive intuitive explanations.
> > > > > > > >
> > > > > > > > Once again, thank you for your suggestions and for helping us improve our paper!

---

### Official Review · Reviewer_bB1m · 2024-10-28

**Soundness:** 3
**Presentation:** 3
**Contribution:** 3
**Rating:** 6
**Confidence:** 3

**Summary:**

This paper is concerned about text embeddings. This paper trickily discovers an interesting phenomenon where text embeddings of lengthy texts tend to cluster very close to each other and thus the performance of text embeddings is largely degraded. In this paper, this phenomenon is termed length collapse. This paper dedicates a lot efforts to connecting this phenomenon to theory through the lens of fourier transform. Base on the theoretical insight, this paper proposes a very simple, intuitive, yet effective method that imposes a temperature scale to the attention mechanism.

**Strengths:**

- The length collapse phenomenon is as far as I know not that new. However, this paper might be the very first one that shed theoretical insights on this phenomenon.
- The proposed tempscale is very intuitive and easy to understand.

**Weaknesses:**

- The experimental results though showcase that tempscale is performant, yet the performance gains over baselines are very marginal.
- I am wondering whether long-context llms still suffer this length collapse, and perhaps furthers discussions on this direction would be more exciting and expected.

**Questions:**

N/A

---

> ### Author Response · Authors · 2024-11-19
> **Author Comment: Performance gain on MTEB**
>
> We are grateful for the reviewer's recognition and insightful comments. We will address the questions below.
>
> ### W1
> - The experimental results though showcase that tempscale is performant, yet the performance gains over baselines are very marginal.
>
> Thank you for your insightful feedback. We proposed this method, along with corresponding experimental results, to verify the validity of our hypothesis: **that the low-pass filtering effect of attention, which intensifies with increasing sequence length, leads to length collapse**. This analysis guides us in mitigating the filtering effect by applying TempScale. The experimental results also validate the correctness of the analysis process.
>
> Secondly, on the MTEB benchmark, even a 1% improvement is significant [1]. For example, the bge-multilingual-gemma2 model currently ranks 7th on the English leaderboard with a score of 69.88. With a 1% increase, it would achieve a score of 70.58, surpassing SFR-Embedding-2_R to rank 4th. This improvement is notable as it requires no fine-tuning and can be achieved through post-processing.
>
> Additionally, for simplicity in our experimental tables, we set the same temperature for each model, which resulted in only marginal improvements across all tasks. However, assigning different temperatures for each task could better highlight the advantages of TempScale. The specific experimental results are as follows (The best-performing result of tau from {0.95, 0.9, 0.8, 0.7, 0.5} is selected as the value for "TempScale"):
>
> | Setting| Reranking | Summarization | Classification | Clustering | LongEmbd Retrival |STS| Beir Retrival |avg|
> |:-:|:-:|:-:|:-:|:-:|:-:|:-:|:-:|:-:|
> |ANCE| 49.09 | 29.58 |55.27 |33.04 | 34.02 | 66.32 | 36.87 | 43.45 |
> | +TempScale | 49.25 | 29.59 |55.44 |33.28 | 34.07 | 66.47 | 36.93 | 43.57 |
> | Relative Delta (%) |0.32 |0.06 |0.29|0.73|0.17 |0.22 |0.16 |**0.28** |
> | GTR| 54.23 | 29.67 |55.10 |38.65 | 37.33 | 70.11 | 44.98 | 47.15 |
> | +TempScale | 54.22 | 29.83 |55.59 |39.52 | 37.33 | 70.26 | 45.61 | 47.48 |
> | Relative Delta (%) | -0.01 |0.54 |0.88|2.26|0.01 |0.21 |1.41 |**0.76** |
> |GIST| 58.55 | 31.14 |64.75 |44.77 | 38.21 | 75.61 | 52.77 | 52.26 |
> | +TempScale | 58.60 | 32.17 |65.12 |44.66 | 38.35 | 75.61 | 53.41 | 52.56 |
> | Relative Delta (%) |0.08 |3.31 |0.56|-0.25 |0.36 | -0.01 |1.21 |**0.75** |
> | BGE| 58.87 | 31.03 |64.79 |45.80 | 37.46 | 75.88 | 55.29 | 52.73 |
> | +TempScale | 58.97 | 31.87 |64.91 |45.79 | 38.73 | 75.68 | 56.00 | 53.14 |
> | Relative Delta (%) |0.17 |2.71 |0.19|-0.01 |3.41 | -0.26 |1.29 |**1.07** |
> | E5-4K | 53.12 | 30.58 |61.72 |38.82 | 56.01 | 71.77 | 47.22 | 51.32 |
> | +TempScale | 53.47 | 31.26 |62.15 |40.62 | 56.88 | 72.17 | 47.01 | 51.94 |
> |Relative Delta (%) |0.65 |2.25 |0.70|4.62|1.56 |0.55 | -0.44 |**1.41** |
> | Avg. Improvement (%) |0.24 |1.77 |0.53|1.47|1.10 |0.14 |0.72 |0.85 |
>
> By setting different temperatures for different tasks, we can further enhance the performance gains brought by TempScale. Remarkably, we also found that as the base capability and window size of the model increase, the improvements from TempScale become more pronounced, ranging from 0.28% on ANCE to 1.41% on E5-4K. This indicates that our method has the potential to deliver even better results as models continue to advance.
>
> Additionally, as shown in Figure 8 and Figure 9 of the paper, we can set different temperatures for texts of varying lengths to achieve greater performance gains.

---

> > ### Author Response · Authors · 2024-11-19
> > **Author Comment: Long-Context LLMs still suffer length collapse**
> >
> > ### W2
> > - I am wondering whether long-context llms still suffer this length collapse, and perhaps furthers discussions on this direction would be more exciting and expected.
> >
> > Thank you for your suggestion. **To explore whether length collapse also occurs in long-context LLMs**, we selected three widely used LLM-based embedding models from the MTEB benchmark—bge-multilingual-gemma2[2], NV-Embed-v2[3], and e5-mistral-7b-instruct[4]—all of which rank within the top 20 on the MTEB leaderboard. We conducted experiments using three commonly used long-text datasets, including LongRAG[5] and LongAlpaca-12k[6] from HuggingFace, which were selected based on relevance to the keyword "long" and chosen for their evenly distributed text lengths. Specifically, we analyzed shifts in embedding space across different text length intervals by calculating the average Euclidean distance of each embedding from the central embedding (the mean of all embeddings) and computing cosine similarity between each pair of embeddings as follows.
> >
> > #### Euclidean distance
> > |Dataset |Model| 0-1000 | 1000-2000 | 2000-3000 |
> > |:-:|:-:|:-:|:-:|:-:|
> > | LongRAG/nq_corpus| bge-multilingual-gemma2 |0.94|0.92 |0.91 |
> > || NV-Embed-v2 |0.98|0.91 |0.89 |
> > ||e5-mistral-7b-instruct |0.68|0.64 |0.64 |
> > | LongRAG/hotpot_qa_corpus | bge-multilingual-gemma2 |0.93|0.88 |0.87 |
> > || NV-Embed-v2 |0.95|0.84 |0.79 |
> > ||e5-mistral-7b-instruct |0.70|0.61 |0.57 |
> > |LongAlpaca-12k| bge-multilingual-gemma2 |0.71|0.48 |0.49 |
> > || NV-Embed-v2 |0.89|0.86 |0.87 |
> > ||e5-mistral-7b-instruct |0.69|0.53 |0.53 |
> >
> > #### Pair Cosine Similarity
> > |Dataset |Model| 0-1000 | 1000-2000 | 2000-3000 |
> > |:-:|:-:|:-:|:-:|:-:|
> > | LongRAG/nq_corpus| bge-multilingual-gemma2 |0.39|0.42 |0.45 |
> > || NV-Embed-v2 |0.32|0.47 |0.59 |
> > ||e5-mistral-7b-instruct |0.75|0.77 |0.78 |
> > | LongRAG/hotpot_qa_corpus | bge-multilingual-gemma2 |0.51|0.52 |0.59 |
> > || NV-Embed-v2 |0.48|0.59 |0.69 |
> > ||e5-mistral-7b-instruct |0.75|0.81 |0.85 |
> > |LongAlpaca-12k| bge-multilingual-gemma2 |0.76|0.90 |0.89 |
> > || NV-Embed-v2 |0.48|0.65 |0.68 |
> > ||e5-mistral-7b-instruct |0.78|0.87 |0.87 |
> >
> > The experimental results show that as text length increases, the embeddings from different LLM-based embedding models exhibit a gradual convergence trend, with the average Euclidean distance between embeddings decreasing and pairwise cosine similarity increasing. This indicates that even mainstream long-context LLM embedding models tend to experience embedding convergence (Length Collapse) and reduced distinctiveness in long-text processing due to the low-pass filtering effect.
> >
> > Thank you very much for your detailed review and valuable comments on our paper. Your suggestions have provided important perspectives that helped us improve the content of our manuscript.
> >
> > [1] Improving Text Embeddings with Large Language Models. ACL2024.
> >
> > [2] BGE M3-Embedding: Multi-Lingual, Multi-Functionality, Multi-Granularity Text Embeddings Through Self-Knowledge Distillation. ACL2024.
> >
> > [3] NV-Embed: Improved Techniques for Training LLMs as Generalist Embedding Models. arXiv2024.
> >
> > [4] Text Embeddings by Weakly-Supervised Contrastive Pre-training. arXiv2022.
> >
> > [5] LongRAG: Enhancing Retrieval-Augmented Generation with Long-context LLMs. arXiv2024.
> >
> > [6] LongLoRA: Efficient Fine-tuning of Long-Context Large Language Models. ICLR2024.

---

> > > ### Author Response · Authors · 2024-11-27
> > > **A kind reminder**
> > >
> > > Dear Reviewer,
> > >
> > > First of all, we would like to sincerely thank you for taking the time to review our paper and provide valuable feedback. Your comments have been incredibly helpful in improving the content of the paper.
> > >
> > > As we are currently in the rebuttal stage, we would kindly like to remind you that if you have any further suggestions or feedback on our response, we would greatly appreciate it if you could share them by the 27th. After this date, we will no longer be able to make modifications to the PDF based on the reviewers' comments. Your continued guidance is crucial for us to refine the paper.
> > >
> > > Once again, thank you for your hard work and support, and we look forward to your valuable response.
> > >
> > > Best regards,
> > >
> > > Authors

---

> > > > ### Author Response · Authors · 2024-12-02
> > > > **A kind reminder**
> > > >
> > > > Dear Reviewer **bB1m**,
> > > >
> > > > We sincerely appreciate your valuable feedback on our manuscript. In response to your suggestions, we have enhanced the revised manuscript with additional experiments and analyses.
> > > >
> > > > As the rebuttal period concludes in less than two days, we hope our efforts align with your expectations. If you find our response satisfactory, we would be grateful if you could consider revising the score.
> > > >
> > > > Thank you once again for your insightful guidance.
> > > >
> > > > Warm regards,
> > > >
> > > > Authors

---

### Official Review · Reviewer_7iH8 · 2024-11-04

**Soundness:** 3
**Presentation:** 2
**Contribution:** 3
**Rating:** 6
**Confidence:** 4

**Summary:**

The paper utilizes Fourier Transform and low-pass filtering as the theoretical framework to analyze the phenomenon of length collapse in embedding models. Based on this analysis, the authors propose TempScale, which introduces a temperature parameter in the softmax function to mitigate the collapse phenomenon. The effectiveness of TempScale is demonstrated through its performance on MTEB and LongEmbed. Generally, I highly appreciate the theoretical framework adopted by the authors and believe that the identified length collapse phenomenon is noteworthy. However, I find the comparison of baseline methods in the experiment section somewhat lacking (see weaknesses). I am very much looking forward to seeing this issue addressed.

**Strengths:**

- The theoretical framework (Fourier Transform and low-pass filtering) is sound, and the identified length-induced embedding collapse phenomenon is noteworthy.

- The experimental section discusses the impact of the length collapse phenomenon on various tasks in MTEB and LongEmbed, which is commendable.

**Weaknesses:**

- The proposed method, $softmax(\frac{1}{\tau\sqrt{d}}QK^T)V$, lacks comparison to previous works (although not specifically proposed for embedding models), including:
  1) $softmax(\frac{logn}{\sqrt{d}}QK^T)V$, as proposed in “Overcoming a Theoretical Limitation of Self-Attention” (ACL 2022);
  2) $softmax(\frac{1}{\tau\sqrt{d}}QK^T)V$, $\tau=0.1ln(n)+1$, as proposed in “YaRN: Efficient Context Window Extension of Large Language Models” (ICLR 2024).

  For both methods, $n$ refers to the input sequence length. I suggest that the authors include a discussion on the differences between TempScale and these two methods. The authors could provide a brief comparison of the theoretical motivations behind TempScale and the two mentioned methods; or include an ablation study comparing the performance of TempScale against these two methods on a subset of the MTEB or LongEmbed tasks; or discuss any potential advantages or limitations of TempScale compared to these existing approaches, particularly in the context of embedding models, etc. Have this issue addressed would positively influence my evaluation.

**Questions:**

None

---

> ### Author Response · Authors · 2024-11-19
> **Author Comment: Scenario Comparison**
>
> We are grateful for the reviewer's recognition and insightful comments. We address the questions below.
>
> >  - The proposed method, $\text{softmax}\left(\frac{1}{\tau \sqrt{d}} QK^T \right) V$, lacks comparison to previous works (although not specifically proposed for embedding models), including:
> >    1. $\text{softmax}\left(\frac{\text{logn}}{\sqrt{d}} QK^T \right) V$, as proposed in "Overcoming a Theoretical Limitation of Self-Attention" (ACL 2022);
> >    2. $\text{softmax}\left(\frac{1}{\tau \sqrt{d}} QK^T \right) V$, $\tau = 0.1 \ln(n) + 1$, as proposed in "YaRN: Efficient Context Window Extension of Large Language Models" (ICLR 2024).
> >
> >     For both methods, $n$ refers to the input sequence length. I suggest that the authors include a discussion on the differences between TempScale and these two methods. The authors could provide a brief comparison of the theoretical motivations behind TempScale and the two mentioned methods; or include an ablation study comparing the performance of TempScale against these two methods on a subset of the MTEB or LongEmbed tasks; or discuss any potential advantages or limitations of TempScale compared to these existing approaches, particularly in the context of embedding models, etc. Have this issue addressed would positively influence my evaluation.
>
> Thank you for your insightful suggestions; the methods you recommended significantly enrich the conclusions of our paper. **The primary aim of our work in developing TempScale was to validate our theoretical framework, which proposes that length collapse arises due to low-pass filtering.** This theoretical analysis was central to our study and guided the development of TempScale as a practical solution. In other papers, there may be similar approaches as you provided, but they may differ in many aspects, such as the theoretical starting points and the problems being addressed.
>
> Next, we will compare the provided methods with TempScale from four perspectives: **1.the scenarios each method addresses**, **2.the theoretical foundations and motivation**, **3.the experimental results in embedding contexts**, and **4.the analysis of these results**. For clarity, we will refer to the scaling methods in your referenced paper as Method1 and Method2.
>
> ### 1. Scenario Comparison:
> Although Method1, Method2, and TempScale appear similar in form, these three methods address different problems in three scenarios.
>
> - Method1 addresses the task of determining if the first string in a sequence of binary (0 and 1) strings is a '1'.
> - Method2 focuses on expanding the context window of large language models (LLMs) without retraining. **After reviewing the original YaRN paper, we noted slight differences in notation and formulas compared to those you provided**. In YaRN, the scaling formula for the attention matrix is given as $\text{softmax}\left(\frac{1}{\tau \sqrt{d}} QK^T \right) V$, where $\frac{1}{\tau} = 0.1 \ln{s} + 1$, and $s = \frac{L^{\prime}}{L}$, with $L^{\prime}$ representing the extended context window length and $L$ the original context window length.
> - TempScale addresses how to mitigate the length collapse effect in embedding models on long texts, improving performance without retraining the model.

---

> > ### Author Response · Authors · 2024-11-19
> > **Author Comment: Theoretical Comparison**
> >
> > ### 2. Theoretical Comparison:
> > Method1 offers a solution using a specially designed Transformer example, but its motivation is limited by the specificity of this example. In contrast, Method2 introduces an empirically derived scaling formula, though the YaRN paper doesn’t explain the reasoning behind it; we attempt to provide this derivation. TempScale, however, is based on a rigorous low-pass filtering analysis, giving it a strong theoretical foundation and an intuitive explanation.
> >
> >    - **Method1**: This method involves building a Transformer to identify if the first string in a sequence of binary strings is "1". The authors observed that, as the sequence length $n$ increased, model performance dropped for sequences longer than the training length. They found that applying Method1 to the attention matrix mitigated performance declines on unseen lengths. However, this solution does not provide a theoretical basis, serving instead as an empirical fix within a specific model and task framework.
> >
> >    - **Method2**: While no specific motivation is mentioned for Method2, its scaling factor was empirically determined using the LLaMA2 model. After reviewing the original YaRN paper and OpenReview sources, the reasoning behind the formula choice remains unclear. Here, I attempt to derive a mathematical rationale for Method2. Assuming the attention matrix has the form $\text{softmax}\left(\frac{\tau}{\sqrt{d}} QK^T \right) V$, each element $a_{i,j}$ of the matrix can be interpreted as a conditional distribution, where the entropy of the attention at position $i$ can be defined as $\mathcal{H}_i=-\sum _ {j=1}^{n} a _ {i,j} \log a _ {i,j}$ . Method2’s extension principle combines "high-frequency extrapolation" with "low-frequency interpolation." For interpolation, we aim for consistent entropy $\mathcal{H} _ i$ at position $i$ before and after RoPE interpolation, requiring careful design of $\tau$ in the attention matrix. We can calculate $\mathcal{H} _ i$ after incorporating $\tau$ as follows:
> >
> >      $$
> >      \mathcal{H} _ {i} = \log\sum _ {j=1}^{n} e^{\tau q _ {i}\cdot k _ {j}} - \frac{\sum _ {j=1}^{n} e^{\tau q _ {i}\cdot k _ {j}}(\tau q _ {i}\cdot k _ {j})}{\sum _ {j=1}^{n} e^{\tau q _ {i}\cdot k _ {j}}},
> >      $$
> >
> >      Assuming $q$ and $k$ are normally distributed with mean 0 and variance 1, we approximate:
> >
> >      $$
> >      \sum _ {j=1}^{n} e^{\tau q _ i \cdot k _ j} \approx n \mathbb{E} _ j[e^{\tau q _ i \cdot k _ j}],
> >      $$
> >
> >      leading to
> >
> >      $$
> >      \mathcal{H} _ {i} \approx \log n + \log \mathbb{E} _ {j}\left[e^{\tau q _ {i} \cdot k _ {j}}\right] - \frac{\tau \mathbb{E} _ {j}\left[e^{\tau q _ {i} \cdot k _ {j}} (q _ {i} \cdot k _ {j})\right]}{\mathbb{E} _ {j}\left[e^{\tau q _ {i} \cdot k _ {j}}\right]}.
> >      $$
> >
> >      Substituting $q _ i \cdot k _ j$ with $d \cos \theta$, we obtain
> >
> >      $$
> >      \mathcal{H} _ {i} \approx \log n + \log \mathbb{E} _ {\theta}\left[e^{\tau d \cos \theta}\right] - \frac{\tau d \mathbb{E} _ {\theta}\left[e^{\tau d \cos \theta} \cos \theta\right]}{\mathbb{E} _ {\theta}\left[e^{\tau d \cos \theta}\right]}.
> >      $$
> >
> >      Applying Laplace approximation yields
> >
> >      $$
> >      \mathcal{H} _ i \approx \log n - 0.24\tau d + \mathcal{O}(1).
> >      $$
> >
> >      Since we aim for consistent entropy $\mathcal{H} _ i$ before and after RoPE interpolation, we set
> >
> >      $$
> >      \log n - 0.24 d + \mathcal{O}(1) = \log sn -0.24\tau d + \mathcal{O}(1),
> >      $$
> >
> >      leading to $\tau \approx \frac{\log s}{0.24} + 1$.
> >
> >    - **TempScale**: The main motivation of this method is to introduce a temperature that reduces the low-pass filtering effect in the attention matrix, allowing the final embedding to retain more high-frequency information, thus providing a more "contextual" representation. This method starts from the attention mechanism itself and does not address length generalization. Instead, it focuses on refining attention within the model’s training window.
> >
> > Method2’s proof does not consider low-pass filtering because it is tailored to the specific scenario Method2 addresses—primarily, how to enable the original position embeddings from training to function effectively on longer texts. The solution includes interpolation, leading to the intuitive idea that the “information aggregation” capability of the attention matrix should remain consistent before and after interpolation. In the proof, “information aggregation” is measured by the entropy of the attention distribution. Moreover, we can also view Method2 from a low-pass filtering perspective: position embeddings originally help the attention matrix to better select information, which preserves high-frequency details in the sequence. When using the original $n$ position embeddings to model a sequence of length $sn$, scaling is needed to focus the model more on specific tokens, thereby retaining more high-frequency information.

---

> > > ### Author Response · Authors · 2024-11-19
> > > **Author Comment: Experimental Results Comparison**
> > >
> > > ### 3. Experimental Results Comparison:
> > > To effectively compare the performance of the above methods, we tested them on the MTEB benchmark. Since we are not modifying the context window length, we adapted Method2 as $\text{softmax}\left(\frac{\tau}{\sqrt{d}} QK^T \right) V$, with $\tau = \lambda \log n + 1$. We explored values for $\lambda$ in the range $\{0.0001, 0.001, 0.01, 0.1, 1\}$. This setup is reasonable, as our findings indicate that longer texts generally benefit from a smaller temperature scale. The experimental results are as follows:
> > >
> > > |Model | Reranking | Summarization | Classification | Clustering | LongEmbd Retrival |STS| Beir Retrival |avg|
> > > |:-:|:-:|:-:|:-:|:-:|:-:|:-:|:-:|:-:|
> > > |ANCE| 49.09 | 29.58 |55.27 |33.04 | 34.02 | 66.32 | 36.87 | 43.45 |
> > > |+Method1| 41.80 | 29.21 |48.21 |20.72 |6.23 | 53.40 |7.74 | 29.61 |
> > > | +Method2($\lambda=0.0001$) | 49.09 | 29.58 |55.62 |33.01 | 34.02 | 66.32 | 36.86 | 43.50 |
> > > |+Method2($\lambda=0.001$) | 49.08 | 29.57 |55.62 |33.05 | 33.96 | 66.32 | 36.87 | 43.49 |
> > > |+Method2($\lambda=0.01$)| 49.10 | 29.31 |55.65 |32.95 | 33.92 | 66.27 | 36.87 | 43.44 |
> > > | +Method2($\lambda=0.1$)| 48.95 | 29.30 |55.45 |32.65 | 32.31 | 65.72 | 35.77 | 42.88 |
> > > |+Method2($\lambda=1$) | 45.71 | 29.28 |52.80 |26.16 | 10.59 | 60.15 | 13.34 | 34.00 |
> > > | GTR| 54.23 | 29.67 |55.10 |38.65 | 37.33 | 70.11 | 44.98 | 47.15 |
> > > |+Method1| 38.12 | 28.44 |40.34 |14.46 |3.61 | 47.09 |0.82 | 24.70 |
> > > | +Method2($\lambda=0.0001$) | 54.23 | 29.68 |55.06 |38.30 | 37.42 | 70.11 | 44.98 | 47.11 |
> > > |+Method2($\lambda=0.001$) | 54.23 | 29.71 |55.08 |38.29 | 37.56 | 70.11 | 44.97 | 47.13 |
> > > |+Method2($\lambda=0.01$)| 54.23 | 29.71 |55.04 |38.63 | 37.39 | 70.06 | 45.00 | 47.15 |
> > > | +Method2($\lambda=0.1$)| 54.12 | 29.34 |54.86 |34.07 | 36.53 | 69.86 | 43.06 | 45.98 |
> > > |+Method2($\lambda=1$) | 55.19 | 29.00 |47.27 |19.06 |3.99 | 61.87 |1.46 | 31.12 |
> > > |GIST| 58.55 | 31.14 |64.75 |44.77 | 38.21 | 75.61 | 52.77 | 52.26 |
> > > |+Method1| 58.64 | 28.26 |51.05 |28.11 |5.42 | 62.43 |7.51 | 34.49 |
> > > | +Method2($\lambda=0.0001$) | 58.55 | 31.14 |64.26 |44.75 | 38.21 | 75.61 | 52.78 | 52.19 |
> > > |+Method2($\lambda=0.001$) | 58.55 | 31.12 |64.28 |44.75 | 38.20 | 75.61 | 52.78 | 52.18 |
> > > |+Method2($\lambda=0.01$)| 58.53 | 31.26 |64.24 |44.78 | 38.05 | 75.60 | 52.79 | 52.18 |
> > > | +Method2($\lambda=0.1$)| 58.45 | 30.36 |63.86 |44.62 | 37.34 | 75.31 | 52.10 | 51.72 |
> > > |+Method2($\lambda=1$) | 55.75 | 26.76 |58.79 |36.95 | 11.85 | 69.00 | 23.59 | 40.38 |
> > > | BGE| 58.87 | 31.03 |64.79 |45.80 | 37.46 | 75.88 | 55.29 | 52.73 |
> > > |+Method1| 37.78 | 29.17 |37.80 |14.36 |2.10 | 43.72 |0.84 | 23.68 |
> > > | +Method2($\lambda=0.0001$) | 58.86 | 31.04 |64.78 |45.77 | 37.45 | 75.88 | 55.29 | 52.72 |
> > > |+Method2($\lambda=0.001$) | 58.86 | 30.96 |64.78 |45.75 | 37.47 | 75.87 | 55.28 | 52.71 |
> > > |+Method2($\lambda=0.01$)| 58.85 | 31.02 |64.73 |45.61 | 37.26 | 75.86 | 55.22 | 52.65 |
> > > | +Method2($\lambda=0.1$)| 58.86 | 30.90 |64.51 |45.19 | 36.55 | 75.55 | 54.96 | 52.36 |
> > > |+Method2($\lambda=1$) | 52.65 | 29.09 |50.51 |25.50 |2.36 | 64.20 |2.14 | 32.35 |
> > > |E5-4K | 53.12 | 30.58 |61.72 |41.01 | 56.01 | 71.77 | 47.22 | 51.63 |
> > > |+Method1| 40.90 | 24.11 |42.85 |13.64 |3.24 | 41.62 |1.05 | 23.92 |
> > > | +Method2($\lambda=0.0001$) | 53.12 | 30.57 |61.78 |40.77 | 56.01 | 71.77 | 47.22 | 51.61 |
> > > |+Method2($\lambda=0.001$) | 53.11 | 30.55 |61.78 |40.75 | 55.98 | 71.77 | 47.22 | 51.59 |
> > > |+Method2($\lambda=0.01$)| 53.07 | 30.42 |61.73 |40.61 | 55.53 | 71.71 | 47.21 | 51.47 |
> > > | +Method2($\lambda=0.1$)| 52.64 | 30.20 |61.51 |40.19 | 48.39 | 70.68 | 46.36 | 50.00 |
> > > |+Method2($\lambda=1$) | 45.00 | 29.01 |50.81 |17.99 |2.91 | 57.27 |1.46 | 29.21 |

---

> > > > ### Author Response · Authors · 2024-11-19
> > > > **Author Comment: Experimental Analysis**
> > > >
> > > > ### 4. Experimental Analysis:
> > > > The experimental results indicate that neither Method1 nor Method2 effectively adapt to the embedding model scenario. (Although Method2 shows slight improvement when the $\lambda$ value is small, in this case, Method 2 degenerates into TempScale.) Some potential reasons are as follows:
> > > >
> > > > - A plausible explanation is that while both methods perform finer scaling on the attention matrix, applying different temperature adjustments across varying text lengths may lead to embeddings from different lengths **falling into distinct distributions**, which is unfavorable for downstream tasks.
> > > >
> > > > - In generation tasks, Method1 and Method2 succeed likely because of differences in output requirements. For these tasks, the model outputs a probability distribution and samples from it. Minor perturbations generally don’t affect the token output significantly; even if one sequence’s token distribution changes, **it doesn't impact the output of other sequences**. In contrast, any substantial change in a single embedding for an embedding model can directly affect the entire downstream performance. For instance, in classification tasks, a classifier model relies on embeddings as input, and in retrieval tasks, a change in embedding impacts document ranking.
> > > >
> > > > Overall, for embedding tasks, simply applying a single temperature adjustment better maintains the overall embedding distribution, helping mitigate length collapse and achieve better results across various downstream applications.
> > > >
> > > > Thank you for your thorough feedback and constructive comments. Your questions and suggestions not only prompted us to reflect deeply on our results but also provided useful inspiration for our future research directions.

---

> > > > > ### Comment · Reviewer_7iH8 · 2024-11-23
> > > > > **Reply to the Authors**
> > > > >
> > > > > Dear authors,
> > > > >
> > > > > Thank you for your response. I have raised my rating to 6.
> > > > >
> > > > > However, I think it's necessary to include the comparison with Method1 and Method2 into the paper. (I have not found them in the current draft, correct me if I'm wrong).
> > > > >
> > > > > Additionally, I really hope, though not compulsory, you could include the following baseline: $softmax(\frac{log(s)}{\sqrt{d}}QK^T)V$, where $s=max(1,log_{L_o}^{n})$. Here, n is the input sequence length, and $L_o$ is the original context length supported by the model, such as 512. The source of this baseline is https://spaces.ac.cn/archives/8823, which also contains derivations similar to the Theoretical Comparison section in your response.

---

> ### Author Response · Authors · 2024-11-23
> **Thanks for Reviewer Comments**
>
> Thank you for your positive feedback and for raising the score. We truly appreciate your support and helping us to improve our paper!
>
> Firstly, we will soon update the paper to include a comparison with Method1 and Method2, and we will notify you once we have made these revisions.
>
> Secondly, regarding the baseline you provided, our paper focuses on the issue of long texts within the model's training context window, where the length of the input text is $n < L_0$, resulting in $\text{max}(1, \text{lon}_{L_0}^n) = 1$. When $s = 1$, $\log(s) = 0$, which causes the Attention matrix to degrade into an average pooling form, rendering self-attention mechanism ineffective and preventing us from applying this method. If you have other suggestions for baselines, we would be very happy to implement them and incorporate them into the paper, as your valuable suggestions truly help us improve our work.

---

> > ### Comment · Reviewer_7iH8 · 2024-11-24
> > **Reply to the authors.**
> >
> > Dear authors,
> >
> > Thanks for you explanation. I realized that there's a typo in my previous response, where the baseline I intended to express should be  $softmax(\frac{s}{\sqrt{d}}QK^T)V$, where $s=max(1,log_{L_o}^{n})$. But, anyway, since you are focusing on the issue of long texts within the model's training context window, it seems that $s$ will always be $1$. Looking forward to the version with comparison of Method1 & 2 included.

---

> > > ### Author Response · Authors · 2024-11-24
> > > **Thanks for Reviewer Comments**
> > >
> > > Thank you for your clarification and helpful suggestions.
> > >
> > > In the revised version of the paper, we have included a comparison between Method1 & Method2, as you suggested. The details have been added in **Appendix G.3:**  Comparison with Other Similar Long-Text Methods, where we provide an analysis and discussion of the comparison.
> > >
> > > We hope this addition clarifies the points you raised.

---

### Official Review · Reviewer_j2YY · 2024-11-09

**Soundness:** 3
**Presentation:** 3
**Contribution:** 2
**Rating:** 5
**Confidence:** 4

**Summary:**

This paper addresses the ineffective performance of attention-based embedding models with long texts. It begins by highlighting that commonly used models struggle with longer inputs. Utilizing visualization techniques, such as t-SNE plots that demonstrate embeddings clustering near the origin, along with statistical evidence like high similarity among long-text embeddings, the authors illustrate a "Length Collapse" issue. This study analyzes the low-pass filtering effect of the attention mechanism, which contributes to this poor performance. Finally, the authors propose a solution by incorporating a temperature parameter in the softmax function to enhance the model's capacity for handling long texts.

**Strengths:**

1. The paper is logically coherent and well-written, ensuring a smooth reading experience.
2. It presents a rigorous theoretical framework supported by substantial relevant research.
3. Additionally, it provides a cohesive analysis of the "Length Collapse" problem, starting with the introduction of the issue, followed by preliminary analysis, in-depth theoretical exploration, and the proposal of a solution based on research findings.

**Weaknesses:**

1. The MTEB benchmark includes 56 evaluation datasets, but this paper evaluates only 36 of them.
2. The paper attributes its poor performance in modeling long texts to the low-pass filtering effect of the attention mechanism, though I question whether this is the primary cause. My concerns include:
   - In Figure 1b, the t-SNE visualization shows long text embeddings clustering near the origin, but this proximity is in the reduced-dimensional space and doesn't confirm their position in the original embedding space.
   - If low-pass filtering significantly affects long-text modeling, it's puzzling that many large models can effectively handle texts of tens of thousands or even hundreds of thousands of tokens, especially since the long texts in this paper are only a few hundred tokens, which is minor compared to what current models manage.
   - Figure 5 indicates that $\sigma_a$ values for text lengths between 100 and 500 are similar, yet Figure 1a reveals a significant performance drop in this range, further undermining the proposed analysis's relevance to the problem.
   - Lastly, I wonder if the training corpus's text length contributes to Length Collapse. Figure 1c shows high similarity among long texts, which is often seen in embeddings before fine-tuning (e.g., embeddings from backbone models). Thus, a possible reason for the poor performance in modeling long texts could be insufficient training data containing longer texts.

**Questions:**

See above.

---

> ### Author Response · Authors · 2024-11-19
> **Author Comment: MTEB datasets**
>
> We are grateful for the reviewer's recognition and insightful comments. We address each point below.
>
> ### W1
>
> > The MTEB benchmark includes 56 evaluation datasets, but this paper evaluates only 36 of them.
>
> **To validate the correctness of our analysis process**, we **randomly selected** datasets from all tasks within the MTEB benchmark to cover as many datasets and tasks as possible. This strategy enabled us to obtain broadly representative evaluation results within limited resources. The data gaps mainly come from 13 datasets in Beir Retrieval. Regarding this, we considered other factors when selecting datasets:
>
> - **Time Consumption**: When choosing evaluation datasets, we carefully weighed the limitations of computational resources and time costs. Some datasets are challenging to evaluate due to their large size. For example, the MSMARCO dataset in the BEIR benchmark contains approximately 8.84 million documents, and a single evaluation run takes about 5-6 hours. Testing across multiple models with different temperature settings would significantly increase the experimental time. Therefore, we excluded this dataset from our evaluation to ensure rational resource utilization.
>
> - **Length Representation**: For retrieval tasks in BEIR, we specifically selected datasets such as Scifact and NFCorpus, which contain **relatively long documents**, to better observe the model's performance on long-text tasks. For instance, in the test sets, the average document length in Scifact is 1,422.3 characters, and in NFCorpus, it is 1,462.7 characters. This design helps us more effectively verify the advantages of our method in long-text tasks.
>
> To address the reviewers' focus on datasets, we provided additional evaluation results in our rebuttal for other retrieval datasets with long documents, including SICDOS (Avg. char 1161.9), FiQA2018 (Avg. char 760.4), and Touche2020 (Avg. char 1117.4). These results report the performance of TempScale on these datasets, offering a more comprehensive demonstration of our approach's effectiveness. (The best-performing result of tau from {0.95, 0.9, 0.8} is selected as the value for "TempScale")
>
> |Setting|SCIDOCS|FiQA2018|Touche2020|Avg.|
> |--------------------|---------|----------|------------|:-----:|
> |ANCE|13.11|27.11|20.78|20.34|
> |+Tempscale|13.17|28.61|21.32|21.03|
> |RelativeDelta(%)|0.43|5.53|2.57|2.85|
> |GTR|14.00|35.15|25.89|25.01|
> |+Tempscale|14.10|36.56|25.91|25.52|
> |RelativeDelta(%)|0.71|4.03|0.07|1.60|
> |GIST|21.89|39.15|21.19|27.41|
> |+Tempscale|21.90|39.29|21.10|27.43|
> |RelativeDelta(%)|0.03|0.37|-0.44|-0.01|
> |BGE|21.47|39.10|19.55|26.70|
> |+Tempscale|21.43|38.99|19.87|26.76|
> |RelativeDelta(%)|-0.17|-0.28|1.61|0.39|
> |E5-4K|15.15|29.38|9.87|18.14|
> |+Tempscale|15.32|30.43|11.61|19.12|
> |RelativeDelta(%)|1.13|3.58|17.56|**7.42**|
> |Avg.Improv.(%)|0.43|2.65|4.27|2.45|
>
> It can be observed that our method improves model performance on most datasets and models, especially achieving a 7% improvement on the E5 model. This further confirms the effectiveness of our approach for long texts and models with extended context windows.
>
> We understand the importance of comprehensive evaluation, and in future work, we plan to expand the range of experiments to cover more datasets, enhancing the thoroughness of our study as resources permit.

---

> > ### Author Response · Authors · 2024-11-19
> > **Author Comment: t-SNE visualization**
> >
> > ### W2
> >
> > > The paper attributes its poor performance in modeling long texts to the low-pass filtering effect of the attention mechanism, though I question whether this is the primary cause. My concerns include:
> >
> > > - In Figure 1b, the t-SNE visualization shows long text embeddings clustering near the origin, but this proximity is in the reduced-dimensional space and doesn't confirm their position in the original embedding space.
> >
> > Thank you for your attention to our t-SNE visualization. **Our main purpose in using t-SNE was to better illustrate the overall distribution trends of long and short texts on a 2D plane through dimensionality reduction, rather than to obtain precise embedding positions.** We are aware that t-SNE introduces a certain degree of spatial distortion during dimensionality reduction, so the clustering positions should not be directly equated with distances in the original embedding space. Additionally, the trend shown in Figure 1c—where the average pairwise cosine similarity between embeddings of different length intervals and the central embedding (the mean of all embeddings) increases with text length—partially indicates that embeddings for longer texts tend to converge toward the center of all embeddings.
> >
> > Furthermore, to provide a more intuitive explanation of the differences among embeddings of varying lengths, we calculated the average Euclidean distance from the embeddings in different length intervals to the central embedding on the NFCorpus and SciFact datasets.
> >
> > #### NFCorpus
> > |Model|0-100|100-200|200-300|300-400|400-500|
> > |:-----:|:-----:|:-------:|:-------:|:-------:|:-------:|
> > |ANCE|3.03|2.83|2.71|2.69|2.63|
> > |GTR|0.67|0.63|0.60|0.59|0.60|
> > |GIST|0.61|0.61|0.60|0.59|0.59|
> > |BGE|0.67|0.62|0.57|0.56|0.57|
> > |E5-4K|6.37|5.79|5.41|5.26|5.27|
> >
> > #### SciFact
> > |Model|0-100|100-200|200-300|300-400|400-500|
> > |:-----:|:-----:|:-------:|:-------:|:-------:|:-------:|
> > |ANCE|2.94|2.62|2.60|2.74|2.74|
> > |GTR|0.65|0.60|0.59|0.60|0.61|
> > |GIST|0.62|0.62|0.64|0.64|0.65|
> > |BGE|0.67|0.65|0.64|0.63|0.64|
> > |E5-4K|6.35|5.85|5.56|5.49|5.35|
> >
> > The results in the table indicate that as text length increases, embeddings are positioned closer to the central embedding, meaning that their distribution in space becomes increasingly centered around the overall embedding distribution. This leads to the Length Collapse.

---

> > > ### Author Response · Authors · 2024-11-19
> > > **Author Comment: Low-Filter on LLM**
> > >
> > > > - If low-pass filtering significantly affects long-text modeling, it's puzzling that many large models can effectively handle texts of tens of thousands or even hundreds of thousands of tokens, especially since the long texts in this paper are only a few hundred tokens, which is minor compared to what current models manage.
> > >
> > > Thank you for the thorough review. In response to the reviewer’s questions on the generality of the low-pass filtering effect in long-text processing, we provided detailed analysis and additional experimental results in our response.
> > >
> > > **Detailed Analysis:**
> > >
> > > - **Inherent trend**: Firstly, our research focuses on revealing **the trend between text length and distribution in embedding space**. As stated in the paper, **regardless of the embedding model's capability in handling long texts**, it still shows a tendency toward length collapse for longer texts compared to shorter ones.
> > >
> > > - **Structural differences**: Secondly, our embedding model is based on a bidirectional attention mechanism, **while many mainstream LLMs use a unidirectional attention mechanism.** This difference in attention mechanisms may alter the properties of the attention matrix, potentially impacting the low-pass filtering effect described in Theorem 1.
> > >
> > > - **Capability independence**: Additionally, while LLMs do show greater capability in handling long texts, this advantage primarily stems from their advanced positional encoding strategies, unidirectional attention mechanisms, and larger parameter sizes with extensive training data. However, this does not completely shield LLMs from length effects. Even with these improvements, LLMs may still exhibit a tendency toward decreased performance when processing extremely long texts, **though the onset threshold is higher.**
> > >
> > > **Experimental Results:**
> > >
> > > **To explore whether length collapse also occurs in long-context LLMs**, we selected three widely used LLM-based embedding models from the MTEB benchmark—bge-multilingual-gemma2[1], NV-Embed-v2[2], and e5-mistral-7b-instruct[3]—all of which rank within the top 20 on the MTEB leaderboard. We conducted experiments using three commonly used long-text datasets, including LongRAG[4] and LongAlpaca-12k[5] from HuggingFace, which were selected based on relevance to the keyword "long" and chosen for their evenly distributed text lengths. Specifically, we analyzed distribution shifts in embedding space across different text length intervals, calculating the average Euclidean distance of these embeddings from the central embedding and the cosine similarity between each pair of embeddings.
> > >
> > > #### Euclidean distance
> > > |Dataset |Model| 0-1000 | 1000-2000 | 2000-3000 |
> > > |:-:|:-:|:-:|:-:|:-:|
> > > | LongRAG/nq_corpus| bge-multilingual-gemma2 |0.94|0.92 |0.91 |
> > > || NV-Embed-v2 |0.98|0.91 |0.89 |
> > > ||e5-mistral-7b-instruct |0.68|0.64 |0.64 |
> > > | LongRAG/hotpot_qa_corpus | bge-multilingual-gemma2 |0.93|0.88 |0.87 |
> > > || NV-Embed-v2 |0.95|0.84 |0.79 |
> > > ||e5-mistral-7b-instruct |0.70|0.61 |0.57 |
> > > |LongAlpaca-12k| bge-multilingual-gemma2 |0.71|0.48 |0.49 |
> > > || NV-Embed-v2 |0.89|0.86 |0.87 |
> > > ||e5-mistral-7b-instruct |0.69|0.53 |0.53 |
> > >
> > > #### Pair Cosine Similarity
> > > |Dataset |Model| 0-1000 | 1000-2000 | 2000-3000 |
> > > |:-:|:-:|:-:|:-:|:-:|
> > > | LongRAG/nq_corpus| bge-multilingual-gemma2 |0.39|0.42 |0.45 |
> > > || NV-Embed-v2 |0.32|0.47 |0.59 |
> > > ||e5-mistral-7b-instruct |0.75|0.77 |0.78 |
> > > | LongRAG/hotpot_qa_corpus | bge-multilingual-gemma2 |0.51|0.52 |0.59 |
> > > || NV-Embed-v2 |0.48|0.59 |0.69 |
> > > ||e5-mistral-7b-instruct |0.75|0.81 |0.85 |
> > > |LongAlpaca-12k| bge-multilingual-gemma2 |0.76|0.90 |0.89 |
> > > || NV-Embed-v2 |0.48|0.65 |0.68 |
> > > ||e5-mistral-7b-instruct |0.78|0.87 |0.87 |
> > >
> > > The experimental results show that as text length increases, the embeddings from different LLM-based models exhibit a gradual convergence trend, with the average Euclidean distance between embeddings decreasing and pairwise cosine similarity increasing. This indicates that even mainstream LLM models tend to experience embedding convergence and reduced distinctiveness in long-text processing due to the low-pass filtering effect.

---

> > > > ### Author Response · Authors · 2024-11-19
> > > > **Author Comment: sigma value and insufficient training on long data**
> > > >
> > > > > - Figure 5 indicates that $\sigma_a$ values for text lengths between 100 and 500 are similar, yet Figure 1a reveals a significant performance drop in this range, further undermining the proposed analysis's relevance to the problem.
> > > >
> > > > Thank you for your valuable question. As you noted, $\sigma_a$ remains stable between 100-500, preventing further performance degradation from low-pass filtering. However, the length not only affects the final embedding distribution by influencing the low-pass filtering of the self-attention matrix **but is also influenced by the tokens themselves.** As the number of tokens in a sentence increases, its mean tends to approach the center of the distribution, thereby intensifying the similarity between long texts. Specifically, if a dimension $x_i$ of the embedding follows a standard normal distribution, under low-pass filtering, it can be viewed as the weighted average of $n$ independent random variables: $y = \frac{1}{n} \sum_{i=1}^n x_i$. As $n$ increases, the distribution of $y$ tends toward zero, causing embeddings to concentrate at a fixed point in high-dimensional space. This leads to the convergence of embeddings in long texts, as shown in Figure 3. Our method, TempScale, mitigates this by lowering the temperature to focus on specific tokens, enhancing the diversity of the final representation.
> > > >
> > > > > - Lastly, I wonder if the training corpus's text length contributes to Length Collapse. Figure 1c shows high similarity among long texts, which is often seen in embeddings before fine-tuning (e.g., embeddings from backbone models). Thus, a possible reason for the poor performance in modeling long texts could be insufficient training data containing longer texts.
> > > >
> > > > Thank you for your question. Whether the training corpus includes enough long texts is indeed worth exploring. To analyze this quantitatively, we selected two models to measure their length collapse tendencies. **One of the models is an 'enhanced-length version' of the other, having been further trained with more and longer data.** Specifically，we used the models stella-base-zh-v2 [6] and piccolo-base-zh [7] to calculate the average Euclidean distance to the center embedding and pairwise cosine similarity for different text lengths across various datasets. Notably, piccolo-base-zh was pretrained and fine-tuned on extensive Chinese corpora, while stella-base-zh-v2 was further fine-tuned specifically on texts longer than 500 tokens. We also sequentially selected the top three datasets in classification tasks on MTEB—AmazonReviewsMulti, iFlyTek, and Waimai—for analysis. The experimental results are as follows:
> > > >
> > > > #### Euclidean distance
> > > > | Datset | Model | 0-100 | 100-200 | 200-300 |
> > > > |:-:|:-:|:-:|:-:|:-:|
> > > > | AmazonReviewsMulti | stella-base-zh-v2 | 10.02 | 9.20| 8.83|
> > > > ||piccolo-base-zh| 11.37 |10.55|10.17|
> > > > | IFlyTek| stella-base-zh-v2 | 11.34 |10.29|10.03|
> > > > ||piccolo-base-zh| 12.35 |11.41|11.23|
> > > > | Waimai | stella-base-zh-v2 | 10.55 | 8.13| 7.68|
> > > > ||piccolo-base-zh| 11.93 | 9.63| 9.24|
> > > >
> > > > #### Pair Cosine Similarity
> > > > | Datset | Model | 0-100 | 100-200 | 200-300 |
> > > > |:-:|:-:|:-:|:-:|:-:|
> > > > | AmazonReviewsMulti | stella-base-zh-v2 |0.81 | 0.82| 0.84|
> > > > ||piccolo-base-zh|0.78 | 0.80| 0.82|
> > > > | IFlyTek| stella-base-zh-v2 |0.73 | 0.77| 0.78|
> > > > ||piccolo-base-zh|0.71 | 0.76| 0.77|
> > > > | Waimai | stella-base-zh-v2 |0.79 | 0.88| 0.91|
> > > > ||piccolo-base-zh|0.75 | 0.85| 0.89|
> > > >
> > > > The experimental results reveal an unexpected finding: although stella-base-zh-v2 was further fine-tuned on long texts, it exhibited an even more pronounced length collapse in terms of Euclidean distance, with longer texts clustering closer to the center embedding. In terms of pairwise cosine similarity, both models displayed a similar trend of length collapse. These results suggest that fine-tuning with more long texts does not alleviate length collapse and may even intensify it.
> > > >
> > > > We sincerely appreciate your careful review and guidance. Your comments have helped us gain a more comprehensive understanding of the shortcomings in our work, allowing us to improve the quality of the manuscript more effectively. We highly value your suggestions and hope that the revisions meet your expectations.
> > > >
> > > > [1] BGE M3-Embedding: Multi-Lingual, Multi-Functionality, Multi-Granularity Text Embeddings Through Self-Knowledge Distillation. ACL2024.
> > > >
> > > > [2] NV-Embed: Improved Techniques for Training LLMs as Generalist Embedding Models. arXiv2024.
> > > >
> > > > [3] Text Embeddings by Weakly-Supervised Contrastive Pre-training. arXiv2022.
> > > >
> > > > [4] LongRAG: Enhancing Retrieval-Augmented Generation with Long-context LLMs. arXiv2024.
> > > >
> > > > [5] LongLoRA: Efficient Fine-tuning of Long-Context Large Language Models. ICLR2024.
> > > >
> > > > [6] https://hf-mirror.com/infgrad/stella-base-zh-v2
> > > >
> > > > [7] https://hf-mirror.com/sensenova/piccolo-base-zh

---

> ### Author Response · Authors · 2024-11-27
> **A kind reminder**
>
> Dear Reviewer,
>
> First of all, we would like to sincerely thank you for taking the time to review our paper and provide valuable feedback. Your comments have been incredibly helpful in improving the content of the paper.
>
> As we are currently in the rebuttal stage, we would kindly like to remind you that if you have any further suggestions or feedback on our response, we would greatly appreciate it if you could share them by **the 27th**. After this date, we will no longer be able to make modifications to the PDF based on the reviewers' comments. Your continued guidance is crucial for us to refine the paper.
>
> Once again, thank you for your hard work and support, and we look forward to your valuable response.
>
> Best regards,
>
> Authors

---

> > ### Author Response · Authors · 2024-12-02
> > **A kind reminder**
> >
> > Dear Reviewer **j2YY**,
> >
> > We sincerely appreciate your valuable feedback on our manuscript. In response to your suggestions, we have enhanced the revised manuscript with additional experiments and analyses.
> >
> > As the rebuttal period concludes in less than two days, we hope our efforts align with your expectations. If you find our response satisfactory, we would be grateful if you could consider revising the score.
> >
> > Thank you once again for your insightful guidance.
> >
> > Warm regards,
> >
> > Authors

---

### Author Response · Authors · 2024-11-24
**Revised pdf and summary**

Dear reviewers,

Thanks for your hard work, your suggestions really help us to improve our paper. We revised our paper according to your suggestions (**revised parts are marked as blue**) and **re-upload our modified pdf**.

We will summarize our changes as follows:

- We added **more intuitive 3D toy examples to explain how Length Collapse reduces the performance of various downstream tasks and how TempScale functions.** Finally, the experiments confirm the validity of the explanation (see Section 3).

- We added experiments on three **LLM-based embedding models** and observed the Length Collapse phenomenon, as noted by **Reviewers bB1m and j2YY** (see Appendix G).

- We added a discussion on why contrastive learning cannot solve Length Collapse (see Appendix F.1).

- We added comparative experiments, as suggested by the **Reviewers 7iH8 and hoZs**, on other post-processing methods addressing anisotropy (see Appendix F.2) and long-text methods with a similar form to TempScale (see Appendix F.3). We also analyzed why these methods cannot solve Length Collapse.

Finally, we would like to emphasize that the primary contribution of our paper is identifying **the phenomenon of Length Collapse in Transformer-based embedding models.** This phenomenon is widespread across PLM- and LLM-based embedding models and is independent of the model's input context window. Thanks to **Reviewer hoZs's** suggestions, we further conducted an in-depth analysis of how the Length Collapse phenomenon affects the performance of various downstream tasks and thoroughly analyzed the reasons behind it. We also provided an efficient solution TempScale to address this issue. Finally, we respectfully emphasize **the importance of our findings and theoretical analysis for the representation learning community**, as they offer valuable insights into embeddings, which are foundational to neural networks.

If you have any questions, please feel free to reach out before the deadline (Nov. 26); we will respond as promptly as possible.

Best,

Authors

---

### Meta-Review · Area_Chair_w4RF · 2024-12-23

**Metareview:**

The article presents a valuable analysis of the 'length collapse' problem encountered in text embedding models. The authors provide a rigorous theoretical framework, underpinned by substantial relevant research, to understand this phenomenon. Reviewers highlight this framework, utilizing Fourier Transform and low-pass filtering principles, is sound and effectively demonstrates the noteworthy impact of input length on embedding quality, leading to the observed 'length collapse.' Furthermore, the proposed TempScale solution offers a potential promising approach to mitigate this issue and enhance the performance of these models on long texts.

A primary obstacle to this paper achieving its full potential lies in its limited exploration of why Large Language Models (LLMs) do not exhibit the same 'length collapse' phenomenon. In addition, the the proposed TempScale only leads to a marginal improvements on the benchmarks could be another weakness of the submission. More detailed comments and suggestions regarding this issue can be found in the individual reviewer comments.

**Additional Comments On Reviewer Discussion:**

The authors are engaged the discussion period and clarified most of the questions. However, the concern on why LLM doesn't have the same "length collapse" (raised by reviewer j2YY and bB1m) still remains. The authors investigated several LLM-based embedding models, which is different than what reviewer asked. Although it is a different model with different training objectives, if the same analysis applies or should not applies should be discussed.

The new experiment adding long text training data leads to a worse performance is also problematic and cannot be explained.

---

### Decision · Program_Chairs · 2025-01-22

Reject